# New Bounds and a Generalization for Share Conversion for 3-Server PIR

**DOI:** 10.3390/e24040497

**Published:** 2022-04-01

**Authors:** Anat Paskin-Cherniavsky, Olga Nissenbaum

**Affiliations:** Computer Science Department, Ariel University, Ariel 40700, Israel; olga@nissenbaum.ru

**Keywords:** PIR, share conversion, CNF secret sharing, communication complexity

## Abstract

Private Information Retrieval (PIR) protocols, which allow the client to obtain data from servers without revealing its request, have many applications such as anonymous communication, media streaming, blockchain security, advertisement, etc. Multi-server PIR protocols, where the database is replicated among the non-colluding servers, provide high efficiency in the information-theoretic setting. Beimel et al. in CCC 12’ (further referred to as BIKO) put forward a paradigm for constructing multi-server PIR, capturing several previous constructions for k≥3 servers, as well as improving the best-known share complexity for 3-server PIR. A key component there is a share conversion scheme from corresponding linear three-party secret sharing schemes with respect to a certain type of “modified universal” relation. In a useful particular instantiation of the paradigm, they used a share conversion from (2,3)-CNF over $Z_{m}$ to three-additive sharing over $Z_{p}^{\beta }$ for primes $p_{1},p_{2},p$ where $p_{1}\neq p_{2}$ and $m=p_{1}\cdot p_{2}$, and the relation is modified universal relation $C_{S_{m}}$. They reduced the question of the existence of the share conversion for a triple $\left(p_{1},p_{2},p\right)$ to the (in)solvability of a certain linear system over $Z_{p}$, and provided an efficient (in m,logp) construction of such a sharing scheme. Unfortunately, the size of the system is $\Theta (m^{2})$ which entails the infeasibility of a direct solution for big *m*’s in practice. Paskin-Cherniavsky and Schmerler in 2019 proved the existence of the conversion for the case of odd $p_{1}$, $p_{2}$ when $p=p_{1}$, obtaining in this way infinitely many parameters for which the conversion exists, but also for infinitely many of them it remained open. In this work, using some algebraic techniques from the work of Paskin-Cherniavsky and Schmerler, we prove the existence of the conversion for even *m*’s in case p=2 (we computed β in this case) and the absence of the conversion for even *m*’s in case p>2. This does not improve the concrete efficiency of 3-server PIR; however, our result is promising in a broader context of constructing PIR through composition techniques with k≥3 servers, using the relation $C_{S_{m}}$ where *m* has more than two prime divisors. Another our suggestion about 3-server PIR is that it’s possible to achieve a shorter server’s response using the relation $C_{S_{m}^{\prime }}$ for extended $S_{m}^{\prime }⊃S_{m}$. By computer search, in BIKO framework we found several such sets for small *m*’s which result in share conversion from (2,3)-CNF over $Z_{m}$ to 3-additive secret sharing over $Z_{p}^{\beta^{\prime }}$, where $\beta^{\prime }>0$ is several times less than β, which implies several times shorter server’s response. We also suggest that such extended sets $S_{m}^{\prime }$ can result in better PIR due to the potential existence of matching vector families with the higher Vapnik-Chervonenkis dimension.

## 1. Introduction

### 1.1. Private Information Retrieval

Private Information Retrieval (PIR) protocols allow the client to fetch items from the server’s database without disclosing to the server which item was requested. A main challenge in constructing PIR protocols is minimizing the communication complexity. The idea of PIR was introduced by Chor et al. [1], together with the 2-server PIR protocol having the communication complexity $O(n^{1/3})$ for the dataset size *n*. PIR has a wide variety of applications such as anonymous communication [2,3], privacy-preserving media streaming [4], blockchain security [5,6], personalized advertisement [7], location and contact discovery [8,9,10], etc.

The naive approach to PIR is just to make the server send all the items in the database to the client: we stress that PIR cares only about the privacy of the client’s request but not about the privacy of the server. However, it entails a huge communication complexity equal to the size of the database. To shorten the communication complexity and still keep the privacy of the request, there are two main approaches to construct PIR:Historically, the first type of PIR was **a Multi-Server PIR** [1], where the database is replicated for k≥2 non-colluding servers. The client secret-shares its request, and servers locally compute the secret-shared response and send it back to the client. The client recovers the item from the shares of response. Multi-Server PIR protocols, such as [11,12,13,14] are relatively efficient in information-theoretic settings. The requirement of the replicated database kept by the non-colluding parties is restrictive; however, there is a space for such a PIR, preferably in blockchain databases, cloud services, multi-server enterprise ecosystems where a small number of servers (but not all) are likely to be compromised.**Single-Server PIR** protocols work in a computational setting and are built on the basis of homomorphic encryption (FHE, AHE, or SHE). The starting point in single-server PIR is the AHE-based protocol of Kushilevitz and Ostrovski [15]. The early single-server PIR constructions were both computationally and communicationally low efficient, although recently significant progress was made which allow speaking about the practically suitable one-server PIR solutions [16,17,18,19]. For instance, the OnionPIR protocol from SHE [16] achieves a 64 KB request and 128 KB response in the online phase of the protocol (and the same in the offline phase) for all the realistic database sizes.

On the high level, for both approaches, the database is represented as a function (usually, a polynomial) *f* such that for any key *x* and the correspondent value (a record) *y* holds y=f(x). Then, the client has to send the request *x* to the server (servers) in a way that preserves its privacy. For the Single-Server PIR, it means that *x* is sent encrypted, in the Multi-Server paradigm, *x* is secret-shared. Encryption or secret sharing has to be homomorphic so that the server (servers) could compute the function f(x) under the encryption/secret-sharing and send the encrypted or secret-shared response *y* back to the client.

In a 2-server computationally-secure PIR of Gilboa and Ishai [20], the request is shared as a DPF (Distributed Point Function) and has a polylog length. In this case, to compute the shares of the response, only additive operations are needed (DPF sharing is homomorphic in respect of them). However, in the information-theoretic setting, which is the focus of this work, it is still unclear how to construct *efficient* in terms of communication and computation PIR with the secret sharing which is homomorphic in respect of any number of additions and multiplications.

Currently, 3 generations of information-theoretic PIR protocols exist: the first generation originated from the work of Chor et al [1] is based on Reed-Muller codes and have communication complexity $n^{1/\Theta (k)}$, in the second from Beimel et al. [21] they restated some of the previous results in a more arithmetic language, in terms of polynomials, and also considered a certain encoding of the inputs and element-wise secret sharing the encoding, which resulted in $n^{O(k)}$ communication complexity. The third generation from works of Efremenko [11] followed by [22,23,24,25,26], Yekhanin [12], Beimel et al. [13], Dvir and Gopi [14] is based on matching vectors and is the most computationally efficient line of protocols with the complexity $n^{o(1)}$ for database size *n*. In all the 3rd generation schemes, but [14], as was demonstrated by Beimel et al. [13], in fact, the combination of two secret-sharing schemes is utilized, both linear in different groups, and a *share conversion* with respect to some relation, allowing to locally perform some non-linear operation over the shares (apart from the case of the identity relation).

### 1.2. Share Conversion

Suppose that there is some number of parties, each holding a share of a secret *s* which was created by a secret-sharing scheme $Sh_{1}$. **The share conversion** is defined as a process of a local computation performed by those parties based only on their shares and outputting the new shares of the secret $s^{\prime }$ in a different scheme $Sh_{2}$ so that there is some predefined relation between *s* and $s^{\prime }$. A systematic study of share conversion was started by Cramer et al. [27] by considering the case $s=s^{\prime }$ for two arbitrary linear secret sharing systems over different fields.

Let us consider an easy illustrative example: for the function $f\left(x\right)=x_{1}\cdot x_{2}$ over the ring $R^{\prime }$, and for the conversion’s relation $s^{\prime }=s^{2}$, for the input $x=(x_{1},x_{2})$ shared in a linear scheme over the ring *R*, it is possible to compute f(x) in the following circuit: first, according to the linear property of the first scheme, servers locally compute shares of $x_{3}=x_{1}+x_{2}$, then convert shares of $x_{1}$, $x_{2}$ and $x_{3}$ to shares of $x_{1}^{2}$, $x_{2}^{2}$ and $x_{3}^{2}$ over $R^{\prime }$, and finally obtain shares of the response $y=2^{-1}\left(x_{3}^{2}-x_{1}^{2}-x_{2}^{2}\right)$.

This approach, however, leaves room for improvement, as such a conversion usually increases the size of the request and response in PIR, because the conversion is a local operation and therefore it is not a trivial issue: to evaluate the circuit which computes some succinct function f(x) which represents the database, the client forms its request as a proper input to this circuit. In addition, not any circuit is possible to compute within the existing secret sharing and conversion schemes, which means that we are bound to only certain kinds of the circuit families and, depending on the VC-dimension of these families of those certain function families, the proper representation of request might be much larger than the size of the database. Recall that the notion of the VC-dimension was introduced by V. Vapnik and A. Chervonenkis in [28]. Informally, for the boolean function family F, where each f∈F:D→{0,1}, VC-dimension VC(F) is the size of the largest I⊂D such that the set $\left\{{f|}_{I}\;|f\in F\right\}$ of restrictions of functions from F contains all the possible boolean functions over *I*. The higher VC(F) relative to |D|, the more efficient PIR can be built. For a precise definition, see [13].

Using homomorphic properties of secret sharing schemes to perform MPC on shared values is a widely used technique in information-theoretic MPC, initiated by the seminal work of [29]. Indeed, in order to (semi honestly) securely evaluate an algebraic circuit, the parties share their input with Shamir secret sharing. Then, linear combinations can be homomorphically evaluated ‘for free’ via local computation on the shares so that additions can be performed repeatedly any number of times. Multiplications can also be performed, however, multiplying two shared values results in a value shared according to Shamir with the doubled degree. This limits the depth of a circuit computable with (even) 1-privacy if we require that the only communication round will be sending shares for the final reconstruction. This idea transfers to PIR, where inputs come from a single party, so they may also be conveniently preprocessed by it via arbitrarily complex functions (which is not always possible for inputs distributed among multiple parties). For instance, for 3-server PIR, degree-2 polynomials can be locally evaluated if Shamir secret sharing was used. As degree-2 polynomials (over a field) in *n* variables have non-trivially high VC dimension ($n^{2}$), this allows for encoding each input via a vector of $O(2^{n/2})$ entries and using the appropriate share conversion. For *k*-server PIR, different kinds of share conversion may enable us to evaluate a family of shallow circuits that both have high VC dimension and suitable secret sharing with share conversion, allowing us to locally evaluate them. In particular, note that a share conversion for a suitable relation, rather than a function suffice to evaluate circuits of that type.

### 1.3. BIKO Framework

In [13], Beimel, Ishai, Kushilevitz, and Orlov (BIKO) interpret the state of the art 3-server PIR schemes as using share conversion from a (variant) of Shamir secret sharing over a certain ring $R_{m}$ for small composite *m*, applied to circuits stemming from MV codes [30] $\left\{u_{1},\ldots ,u_{h}\right\},\left\{v_{1},\ldots ,u_{h}\right\}$ with a bounded set S∪{0} of $<u_{i},u_{j}>$ values, for some $S\subseteq Z_{m}\backslash \left\{0\right\}$. It has the property that $<u_{i},v_{i}>=0$, while $<u_{j},v_{i}>$ for j≠i is in *S*. We refer to such codes as *S*-bounded MV codes. They manage to get improved complexity of the resulting PIR, by using conversions from CNF secret sharing rather than from Shamir over certain small $R_{m}$, for which a conversion from Shamir for that relation does not exist (the (t,k)-CNF is a threshold secret sharing scheme introduced in [31]; see Section 2.2 for a detailed description). Specifically, they obtained conversions from (2,3)-CNF over $Z_{m}$ to the additive secret sharing scheme over $Z_{p}^{\beta }$ for the following relation $C_{S}=\left\{\left(0,s^{\prime }\right)|s^{\prime }\in Z_{p}^{\beta }\backslash \left\{0\right\}\right\}\cup \left\{(s,0)|s\in S\backslash \{0\}\right\}\cup \left(Z_{m}\backslash \left\{S\cup \{0\}\right\}\right)\times Z_{p}^{\beta }$. They work with the so-called *canonical* set $S=S_{m}=\left\{x\in Z_{m}|\forall i\;x\;\mathrm{is}\;\mathrm{either}\;0\;\mathrm{or}\;1\;\mathrm{mod}p_{i}^{e_{i}}\right\}\backslash \left\{0\right\}$, where $m=\prod_{i=1}^{k}p_{i}^{e_{i}}$ is the decomposition of *m* into prime factors. This is a useful choice, due to the existence of good $S_{m}$-bounded MV codes over composite moduli *m*. Their approach is motivated by the existence of conversions for CNF to additive (roughly, that CNF can be converted to “any” scheme, and any scheme can be converted to additive), they use $Sh_{1}$ as CNF over a certain ring, and $Sh_{2}$ as additive over another ring. This relation (although not a function) suffices to evaluate the required type of circuits, arising from the MV family. There is a potential tradeoff here between the best MV codes that exist over a certain ring *R*, and the size (more generally, the identity) of the set *S* that can be achieved. On a high level:The smaller *S* is, the easier it is to find a suitable share conversion (required to evaluate functions in the circuit family induced by the MV code).The larger *S* is, the easier it is to find an MV code resulting in a family of circuits with high VC dimension. The communication complexity of the resulting PIR decreases with the VC dimension of the set (and eventually, the size of the shallow circuit to evaluate).

The concrete parameters of both constructions used so far for 3-server PIR (in their most efficient variants) follow from the following Theorem 7, and instantiations of it via known constructions of MV codes and share conversion schemes.

On a very high level, these PIR protocols consist of three steps and is shown in Construction 1.
**Construction 1:** BIKO Framework [13]**^1^** Let $f:{\left\{0,1\right\}}^{log(n)}\to \left\{0,1\right\}$ denote the server’s database. The client preprocesses its input $x\in {\left\{0,1\right\}}^{log(n)}$ into a vector $v_{x}\in R^{h}$ for a (constant) ring *R*, where ${\left\{v_{x}\right\}}_{x}$ is a set of vectors of an *S*-bounded MV code. It shares the vector coordinate-wise among the *k* servers via some (2,k)-private secret sharing scheme $Sh_{1}$ (so no single server learns anything about the secret).**^2^** The servers use linear homomorphism properties of $Sh_{1},Sh_{2}$, which are homomorphic over certain finite groups, to locally evaluate (an encoding of) *f* on the shared *v*. More concretely,
$$f\left(v\right)=\sum_{\left\{i|f(i)=1\right\}}f_{i}\left(v\right)$$
where $f_{i}\left(v\right)=<u_{i},v>$. In some more detail, each $<u_{i},v>$ uses linear homomorphism of $Sh_{1}$, then a share conversion from $Sh_{1}$ to $Sh_{2}$ relatively to $C_{S}$, applied to each share of $f_{i}\left(v\right)$, and finally linear homomorphism of $Sh_{2}$ is applied to evaluate $\sum_{i}f_{i}\left(v\right)$ on the resulting shares. The share conversion is required to transform $<v_{i},v>$ for $v_{i}=v$ into a non-zero value, and $<u_{j},v>$ for $v_{j}\neq v$ into 0’s, making the sum non-zero iff. f(v)=1. Then each server sends its share to the client.**^3^** The client recovers the output using linear homomorphism of $Sh_{2}$, and post-processing the value.

The correctness of the scheme is easy to verify.

For a 3-server PIR, Ref. [13] provides the technique for the constructing the conversion (it such a conversion exists) from (2,3)-CNF to the additive secret sharing and obtains results for some special cases. Utilizing the results of Beimel et al., Paskin-Cherniavsky and Schmerler in [32] proved that there is a share conversion from (2,3)-CNF over $Z_{m}$ to 3-additive secret sharing over $Z_{p}$, if $m=p_{1}p_{2}$, for distinct odd primes $p_{1}$ and $p_{2}$, one of which is equal to *p*. Thereby they found infinitely many cases when conversion falling into the BIKO framework exists.

**Theorem** **1**([13,32])**.**
*Let $m=p_{1}\cdot p_{2}$, where $p_{1},p_{2}$ are distinct primes, and p is a prime. Then, there exists a share conversion from (2,3)-CNF to additive over $Z_{p}^{\beta }$ for the relation $C_{S_{m}}$ for some β in the following cases:*
*1* *$p_{1},p_{2}\neq 2$, and $p\in \{p_{1},p_{2}\}$**2.* *$p_{1}=2,p_{2}\in \left\{3,5,7\right\}$ and p=2.*

For other cases of $m=p_{1}\cdot p_{2}$ and *p*, however, the existence of the conversion was neither confirmed nor disproved. The constant β in Theorem 1 seems to grow with *m*, but due to the techniques used, it has not been proven for any infinite family of parameters.

**Remark** **1.**
*However, not all the 3rd generation information-theoretic PIR protocols fall into the BIKO framework. For instance, the work of [14] could be viewed as a certain generalization of it. This beautiful work surprisingly manages to carry over "3rd generation" PIR communication complexity previously achieved for 3 or more servers, to the 2-server setting, resolving a long standing open problem, thereby illustrating the limitations of the BIKO framework, providing evidence that generalizing it in certain directions can be instrumental in the context of PIR. In some more detail, [14]’s PIR has a bilinear, rather than linear reconstruction in $Sh_{2}$, and the step corresponding to share conversion can not be cleanly viewed as a share conversion from $Sh_{1}$ to $Sh_{2}$ according to $C_{S}$ (or in fact any) relation. In particular, the client essentially uses a 2-out-of-3 sharing scheme to make the share conversion work, with himself holding one of the shares.*


### 1.4. Our Contribution


**Obtaining another infinite class of conversions from (2,3)-CNF.**


Following the BIKO framework [13] and utilizing some results of [32], we prove that:

**Theorem** **2**(Main result, informal)**.**
*There exists a share conversion from (2,3)-CNF over $Z_{2q}$ to 3-additive secret-sharing scheme over $Z_{2}^{\left(q-1)(q-2\right)}$ for any odd prime q.**There is no conversion from (2,3)-CNF over $Z_{2q}$ to $Z_{p}^{\beta }$ for any odd primes q and p (including the case q=p) and any β>0.*

In this way, we prove the existence of the conversion for infinitely many cases, and also for infinitely many cases we prove a conversion does not exist. Together with [32] for *m*’s which are products of two primes, it leaves open only the question of the conversion in the case when $m=p_{1}p_{2}$, where $p_{1}$ and $p_{2}$ are both odd and not equal to *p*.

Note also that for considered cases, we managed to compute the parameter β which determines the server’s response size. We prove that β in Theorem 7 is indeed the best for m=6 among $m=p_{1}p_{2}$ where $p_{1}=2$. More concretely, one of our contributions is the precise value of β for share conversion with respect to relation $C_{S_{2q}}$. Previous techniques did not allow to compute β, as they traded generality that could allow computing β for some additional simplicity—using a single row in $M_{≢}$ to understand the rank difference $\beta =rank\left(M_{\equiv ,≢}\right)-rank\left(M_{\equiv }\right)$.


**Computing and improving server reply size.**


Another somewhat surprising observation we made is that we may sometimes increase *S* beyond $S_{m}$ so that a conversion from (2,3)-CNF over $Z_{m}$ to $Z_{q}^{\beta }$ (for the same m,q as before) still exists. This may have two possible implications. A direct implication that we observed experimentally for several values of *m*, is that the rank difference β sometimes goes down, but not all the way to 0. Thus, if the share conversion still exists, as follows from the BIKO technique, β may decrease, leading to the reduced size of the server’s response. We checked this fact for some small *m*’s by computer search and obtained positive results, which is presented in Section 4. Indeed, we obtained smaller β supplementing $S_{m}$ up to $S_{m}^{\prime }$ by additional values. We informally sum the result of the computer search in the following theorem.

**Theorem** **3.**
*There exists a share conversion from (2,3)-CNF over $Z_{m}$ to 3-additive secret-sharing scheme over $Z_{p}^{\beta^{\prime }}$ with respect to the relation $C_{S_{m}^{\prime }}$, refining β, where:*
m=14, p=2, $S_{m}^{\prime }=S_{m}\cup \left\{3\right\}$*and*$S_{m}^{\prime }=S_{m}\cup \left\{5\right\}$, β=30, $\beta^{\prime }=6$;m=15, p=3, $S_{m}^{\prime }=S_{m}\cup \left\{11\right\}$β=24, $\beta^{\prime }=12$;m=21, p=3, $S_{m}^{\prime }=S_{m}\cup \left\{8\right\}$, β=60, $\beta^{\prime }=30$;m=33, p=3, $S_{m}^{\prime }=S_{m}\cup \left\{23\right\}$, β=180, $\beta^{\prime }=90$;m=15, p=5, $S_{m}^{\prime }=S_{m}\cup \left(\mathrm{any}\;\mathrm{non}-\mathrm{empty}\;\mathrm{subset}\;\mathrm{of}\;\left\{4,7,13\right\}\right)$, β=8, $\beta^{\prime }=2$;
*m=35, p=5, $S_{m}^{\prime }=S_{m}\cup \left(\mathrm{any}\;\mathrm{non}-\mathrm{empty}\;\mathrm{subset}\;\mathrm{of}\;\left\{8,22,29\right\}\right)$, β=120, $\beta^{\prime }=30$;*

*m=21, p=7, $S_{m}^{\prime }=S_{m}\cup \left(\mathrm{any}\;\mathrm{non}-\mathrm{empty}\;\mathrm{subset}\;\mathrm{of}\;\left\{4,10,13,16,19\right\}\right)$, β=12, $\beta^{\prime }=2$;*

*m=35, p=7, $S_{m}^{\prime }=S_{m}\cup \left(\mathrm{any}\;\mathrm{non}-\mathrm{empty}\;\mathrm{subset}\;\mathrm{of}\;\left\{6,11,16,26,31\right\}\right)$, β=72, $\beta^{\prime }=12$.*



This result may also be viewed as evidence that canonical sets $S_{m}$ for *m* with a larger number *r* of prime factors may potentially have share conversions for $C_{S_{m}}$ for (significantly) smaller than $2^{r}-1$ number of servers (as we have conversions for $2^{r}-1$ servers but *S* larger than $S_{m}$, where the resulting linear system has much more rows than columns). This direction is interesting to explore, initiating a systematic search for share conversions with server sets as small as possible, resulting in PIR with share complexity polynomial in MV codeword length for *m* which is a factor of *r* primes.

In addition to our two main contributions, **we identify a few minor errors** in [13,32]. Nevertheless, these errors do not affect the correctness of any of their main contributions.
We recalculated some computer search results of [13] (BIKO) as they come in contradiction with the theoretical result of Paskin-Cherniavsky and Schmerler. In particular, [13] showed the absence of the conversion for m=35, p=7, while [32] proved that the conversion for this case exists. In addition, we obtained numerical results for cases m=22, 26, 33 which were not considered in BIKO. Our numerical results given in Section 4 confirm both our theoretical result for $p_{1}=2$ and the conclusion of [32].We corrected some calculation mistakes made in previous work [32]. The corrigenda are shown in Appendix A.

### 1.5. Instantiations of BIKO and Future Directions of Our Work

Almost all third-generation PIR protocols falling in a BIKO framework, utilize the conversion from Shamir secret sharing instead of CNF. The existence of the conversion from Shamir secret sharing scheme implies the existence of conversion from CNF, but not vice versa [13].

The following theorem by V. Grolmusz generalizes a similar instance of the theorem for 3-servers in [13], to put our work in a broader context. It states the size of the MV families depending on the constant *m* which has an impact on the complexity of the PIR protocols based on them.

**Theorem** **4**([30])**.**
*Let $m=\prod_{i=1}^{r}p_{i}$ where the $p_{i}$’s are distinct constant primes, and r>1 is constant. Then there exists an MV code family $C\subseteq Z_{m}^{h}$ of size $\left|C\right|=\exp\left(\frac{c\log^{r}\left(h\right)}{{\log\log(h)}^{\left(r-1\right)}}\right)$ which is $S_{m}$-bounded. Here $c\leq p_{r}^{-r}$, where $p_{r}$ is the largest prime.*

In fact, the construction in Theorem 4 generalizes to any *m* with *r* distinct prime divisors.

Next, we outline some parameters for which suitable share conversions leading to (3rd generation) PIR via the BIKO framework and MV codes from Theorem 4 exist. Note that Theorems 5 and 6 were initially stated in terms of conversion from Shamir secret sharing, but a corresponding conversion from CNF is implied.

**Theorem** **5**([11,26])**.**
*For each r≥2, there exists a number m with r distinct prime divisors $p_{1}\leq \ldots \leq p_{r}$, with $p_{r}\geq 73$, for which there exists a share conversion from $\left(2,3/4\cdot 2^{r}\right)$-CNF over $Z_{m}$ to $3/4\cdot 2^{r}$-additive over $Z_{2}^{\beta }$ for some β<m, and relation $C_{S_{m}}$. Furthermore, such a conversion exists for every m of the form $2^{t}-1$ with r distinct prime divisors, if the number of parties, $3/4\cdot 2^{r}$, is replaced by $2^{r}$.*

In a nutshell, the above result is obtained by [26] via a composition technique applied to [11]’s result for 3-server and $2^{r}$-server PIR. The reduction in the number of parties from $2^{r}$ to $3/4\cdot 2^{r}$ for *m* with *r* prime distinct divisors follows from the (somewhat surprising) 3-party conversion for r=2 and m=73·7.

In [23], the authors found 50 additional such 3-party conversions for $m=p_{1}\cdot p_{2}$ (which need to satisfy a certain condition), leading to further improvements in the number of parties as a function of *r*. Note, that for all *m* found in [23], $p_{r}\geq 73$ are large, so the constant in Theorem 1 grows fast with *r*.

**Theorem** **6**([23])**.**
*For each r≥104, there exists a share conversion from $\left(2,{\left(3/4\right)}^{51}\cdot 2^{r}\right)$-CNF over $Z_{m}$ to ${\left(3/4\right)}^{51}\cdot 2^{r}$-additive over $Z_{2}^{\beta }$ for some β<m, and relation $C_{S_{m}}$. For each r<104, there exists a share conversion from $\left(2,3^{\left\lceil r/2\right\rceil }\right)$-CNF over $Z_{m}$ to $3^{\left\lceil r/2\right\rceil }$-additive over $Z_{2}^{\beta }$ for some β<m, and relation $C_{S_{m}}$.*

Note that for the above instantiations, “descending” from [11], *m* must be odd.

**Theorem** **7**(Implicit in [13])**.**
*Let m∈N, $\left\{0\right\}\subseteq S\subseteq Z_{m}$, and C an S-bounded MV code family $\left\{C_{h}\right\}$ of vectors in $Z_{m}^{h}$. Assume also there exists share conversion from (2,k)-CNF over $Z_{m}$ to $Z_{p}^{\beta }$ for some constant β, for the relation $C_{S}$. Then there exists a k-server PIR family for databases of size $n=\left|C_{h}\right|$ with client’s message of size ⌈hlog(m)⌉ and server’s message of size ⌈βlogp⌉.*

From Theorems 6 and 7 follows

**Corollary** **7.**
*Let r≥3. Then there exists some $m=\prod_{i=1}^{r}p_{i}^{e_{i}}$ where $p_{1}<\ldots <p_{r}$ are primes where $m|(2^{\beta }-1)$ for some β. Then a $3/4\cdot 2^{r}$-server PIR with client communication complexity of $O(2^{2p_{r}\cdot \log^{1/r}\left(n\right)\log\log^{1-1/r}\left(n\right)})$ and (each) server’s communication complexity βlog(p), for some β≤m exists. For r≥104, this improves to ${\left(3/4\right)}^{51}2^{r}$ servers, and $3^{\left\lceil r/2\right\rceil }$ servers for r<104.*


We note that among the known *m*’s in the Corollary above for r=2, $p_{r}\geq 73$ and grows particularly fast with *r* for r≥104 if ${\left(3/4\right)}^{51}2^{r}$ servers (instead of $3/4\cdot 2^{r}$) exist.

Instantiating Theorem 7 with Theorem 4 for MV-code construction, and either Theorem 1, we obtain the best known concrete efficiency of 3-server PIR, with $2^{6\sqrt{\log(n)\log\log(n)}}$ communication complexity. On the other hand, for more than polynomially improved communication complexity and a larger number of servers, the best result is obtained by instantiating the share conversion via Theorem 5.

Our concrete result does not improve communication complexity for 3-server PIR, which is essentially optimal for conversion from $Z_{6}$ by [13] as stated in Theorem 1. However, the technical tools developed may help understand the existence of share conversions for even *m* with a larger number of prime factors, with better the communication complexity of PIR and the larger number of servers. Due to the generality of BIKO’s framework, converting from CNF, one could hopefully get improved efficiency of communication complexity relatively to the number of servers. In particular, as noted above, the instantiation of BIKO as in [11] does not yield PIR protocols with even *m*, and the known values of *m* have large maximal factors and lead to PIR with high constants in the exponent. By a direct corollary from Theorems 4 and 5 similar to Corollary 7, we get a 6-server PIR with communication complexity $O\left(2^{146\cdot \log^{1/3}\left(n\right)\log\log^{2/3}\left(n\right)}\right)$. Using the BIKO framework instantiated Theorem 5—the ‘furthermore’ part, for 8-server PIR we obtain a complexity of $O\left(2^{34\cdot \log^{1/3}\left(n\right)\log\log^{2/3}\left(n\right)}\right)$, by using $m=255=3\cdot 5\cdot 17=2^{8}-1$, and instantiating Theorem 4 with m=255. Thus, as far as we know, no PIR with complexity better than $O\left(2^{146\cdot \log^{1/3}\left(n\right)\log\log^{2/3}\left(n\right)}\right)$ (best known 6-server PIR) exists for 7 servers. We conjecture that a 7-server PIR with much improved constants exists, by using share conversion from (2,7)-CNF with parameters generalizing the conversions we obtained for $m=2\cdot p_{2}$.

**Conjecture** **1.***A share conversion from (2,7)-CNF over $Z_{30}$ to $Z_{2}^{\beta }$ for some constant β exists, implying a 7-server PIR for*$O\left(2^{10\cdot \log^{1/3}\left(n\right)\log\log^{2/3}\left(n\right)}\right)$.

We hope to be able to verify the conjecture more easily by generalizing the insights we have for the existence of a share conversion for $m=2\cdot p_{2}$ to a share conversion to m=2·15 (more generally, for 2·c for some composite *c*), and the fact that in this case of $p_{1}$, the analysis turned out to be rather simple. Another reason to hope we can manage with 7 servers is that $M_{\equiv }$ is in that case, has a form similar to the 3-server case considered in present work (unless, for example, 6-server case). See Section 1.6 for more details.

A broader goal is improving the number of servers one can tolerate for PIR with CC corresponding to MV codes over $Z_{m}$ with *r* prime factors. While [26] show how to achieve ${\left(3/4\right)}^{51}2^{r}$ servers for an infinite number of *r*’s and corresponding *m*’s, and $3^{r/2}$-server PIR for finitely many *r*’s, it would be interesting to improve Theorem 6 to get share conversion for $3^{r/2}$-server PIR for all *r*. Our hope is to devise a composition theorem along the lines of [26], composing ‘gadgets’ of conversions from (2,3)-CNF over $Z_{m}$ for coprime composite *m*’s. As we already have such conversions for infinitely many pairwise coprime *m*’s via Theorem 1, we only need a suitable composition theorem. In fact, it is not hard to show, that if we had conversions for coprime $m_{1},m_{2}$ respectively, both to $Z_{p}^{\beta }$ for the same *p*, say $Z_{2}^{\beta }$, we would obtain the result. In particular, it is strictly easier to prove the existence of conversion from $Z_{p_{1}p_{2}}$ to $Z_{2}^{\beta }$ for some β depending on $p_{1},p_{2}$ for infinitely many coprime $p_{2i+1}\cdot p_{2i+2}$’s (as the 51 known cases based on Mersenne-style primes in [26] are a special case). To summarize, to complete this direction, we only need to find a conversion from (2,3)-CNF over $Z_{m_{i}}$ to $Z_{2}^{\beta_{i}}$ for infinitely many coprimes $m_{i}$’s of the form $m_{i}=p_{1}\cdot p_{2}$ where $p_{1},p_{2}$ are distinct primes. This seems to require only moderate extension on the (linear algebraic) toolbox conversions from (2,3)-CNF that has been laid out in the seminal work of [13] and subsequently in [32].

A more ambitious still direction (which we expect to be more technically involved) is expected to lead to dramatic improvements in the number of servers, bringing it down from exponential to linear in *r*. It relies on the following composition lemma, which is not hard to prove (see full version for details).

**Lemma** **1.**
*Let $m_{1}=2m_{1}^{\prime },\;m_{2}=2m_{2}^{\prime }$, where $m_{1}^{\prime },\;m_{2}^{\prime }>1$ are odd coprime integers. Assume there exists a share conversion from (2,k)-CNF over $Z_{m_{1}}$ to $\left(t_{1},k\right)$-CNF over $Z_{2}^{\beta_{1}}$ for the relation $S_{m_{1}}$ (and an analogous conversion exists for $m_{2}$). Then there exists a share conversion from (2,k)-CNF over $Z_{2m_{1}^{\prime }m_{2}^{\prime }}$ to $\left(t_{1}+t_{2}+1,k\right)$-CNF over $Z_{2}^{max(\beta_{1},\beta_{2})}$ for $C_{S_{2m_{1}^{\prime }m_{2}^{\prime }}}$.*


**Remark** **2.**
*More generally, slightly optimizing parameters, relatively to iteratively applying Lemma 1 for two $m_{i}$’s, for any r≥2, and $m_{1},\ldots ,m_{r}$ as above, we obtain a share conversion from (2,k)-CNF over $Z_{2\prod_{i=1}^{r}m_{i}^{\prime }}$ to $\left(1+\sum_{i=1}^{r}t_{i},k\right)$-CNF over $Z_{2}^{max(\beta_{1},\ldots ,\beta_{r})}$ for the relations $C_{S_{2\prod_{i=1}^{r}m_{i}^{\prime }}}$.*


Assume a conversion generalizing our result from Theorem A1 for 3 servers to more servers, while keeping the conversion to a scheme (t,k)-CNF for sufficiently small *t*. Such a scheme has enough redundancy to support multiplications over the resulting field $F_{2^{\beta }}$ unlike (k,k)-additive, which has none (if needed, the field characteristic 2 may be replaced with some other prime, generalizing Theorem 1 instead). Then we can obtain PIR with linear server complexity k=O(r), using Theorem 4, and applying Lemma 1 r−1 times. More precisely, we have:

**Corollary** **7.**
*Assume there exists a (global) constant t, such that for all sufficiently large k, the following holds. For infinitely many $m_{i}$’s of the form $m_{i}=2p_{i}$ where all $p_{i}$ are odd distinct primes, there exists a share conversion from (2,k)-CNF to (t,k)-CNF over $Z_{2}^{\beta_{i}}$ for the relation $C_{m_{i}}$. Then, for all sufficiently large r, there exists a k=t(r−1)+1-server PIR with communication complexity $2^{O(\log^{1/r}\left(n\right)\log\log^{1-1/r}\left(n\right))}$.*


### 1.6. Our Techniques

As described above, one of the main contributions of [13] was an instantiation of the framework for designing PIR protocols, which reduces the question of the existence of a three-server PIR protocol to the existence of a share conversion for certain parameters $p_{1},p_{2},p$, and certain linear sharing schemes over Abelian rings *R*, $R^{\prime }$ determined by the parameters.

BIKO provides the criteria of the share conversion existence in the case when $m=p_{1}p_{2}$ for distinct primes $p_{1}$ and $p_{2}$ and the set $S_{m}=\left\{s_{1},s_{2},1\right\}$, where $s_{1}\;\mathrm{mod}\;p_{1}=0$, $s_{1}\;\mathrm{mod}\;p_{2}=1$, $s_{2}\;\mathrm{mod}\;p_{1}=1$, $s_{2}\;\mathrm{mod}\;p_{2}=0$. Namely, they prove that for such *m* and $S_{m}$, the share conversion from (2,3)-CNF over $Z_{m}$ to 3-additive scheme over $Z_{p}^{\beta }$ exists if and only if $rank\left(M_{\equiv ,≢}\right)-rank\left(M_{\equiv }\right)=\beta >0$, where the rank is computed over $F_{p}$. The matrices $M_{\equiv }$ and $M_{\equiv ,≢}$ are matrices over $Z_{p}$ with $3m^{2}$ columns and $3m^{2}$ and $4m^{2}$ rows respectively which are constructed from some specific system of equations and inequalities. Beimel et al. in [13] did not provide the general solution for this system; however, they proved existence and nonexistence of the conversion for some special cases.

While the solvability of a system can be verified efficiently for a concrete instance, it does not provide a simple condition for characterizing triples $\left(p_{1},p_{2},p\right)$ for which solutions exist. Moreover, the size of the matrix $M_{\equiv ,≢}$ in this system is $4m^{2}\times 3m^{2}$ which makes the numerical solution for big *m*’s heavy in practice (though asymptotically efficient). Before [32], where the solvability of the system for the case odd primes $p_{1}$ and $p_{2}$, if one of them equals to *p* was proven, even the question of whether an infinite set of such triples exists remained open.

Our concrete goal in this work is to better understand the case of $m=p_{1}p_{2}$, motivated by understanding the technical foundations of the broader problem for *m* which is a product of r>2 distinct primes (see Section 1.5 for details). We proceed using the BIKO characterization above. Concretely, for parameters $m=p_{1}p_{2}$ and *p*, this reduces to calculating the quantity $rank\left(M_{\equiv ,≢}\right)-rank\left(M_{\equiv }\right)=\beta$, where the rank is computed over $Z_{p}$.

In [32], the case $p_{1}=p$ for odd $p_{1}$ and $p_{2}$ was explored. To simplify the technical task, the authors of [32] rely on the observation from [13] that β>0 iff $M_{\equiv }$ does not span $v_{≢}$ for *any* row $v_{≢}$ of $M_{≢}$. Thus, they replace $M_{≢}$ with some $v_{≢}$ as above, and work with that (forgoing the goal of understanding the particular value of β). Then, they proceed by bringing the matrix $M_{\equiv ,≢}$ to a more convenient form by performing a sequence of carefully tailored elimination steps on the rows of the matrix $M_{\equiv ,≢}$. The sequence of eliminations is based on a observing a 3-leveled structure of the matrix of the matrix $M_{\equiv ,≢}$, and working on blocks of decreasing coarseness as the elimination process progresses. It also involves a change of basis at some point, to make the matrix’s structure nicer for understanding. That is, rewriting the matrix so that the set of columns corresponds to a new basis—here we even manage to get fewer vectors in, as it suffices to include a set of vectors which is guaranteed to span $M_{\equiv ,≢}$. However, the resulting matrix after that process remains too complex to check whether β>0 for all parameters. The analysis up to that point (resulting in some matrix $A_{inter}=\left(A_{inter}^{\prime },v_{≢}^{\prime }\right)$ to analyze) is oblivious to the particular parameters except for not looking at even *m* (not because it was particularly hard, but rather out of a decision to limit the scope of the paper at what was already achieved). To obtain their partial result for some of the parameters, the authors then reduce the matrix’s rows modulo a certain vector subspace (formally, multiplied it from the right by a certain square matrix *L* with non-trivial left kernel). Clearly, it holds that if $rank\left(A_{inter}L\right)-rank\left(A^{\prime }L\right)>0$, then $rank\left(A_{inter}\right)-rank\left(A^{\prime }\right)>0$ as well (implying the existence of a share conversion), but not necessarily the other way around. The matrix $A_{inter}L$ turns out to be sufficiently simple to analyze, and for *p* which is either $p_{1}$ or $p_{2}$, the resulting rank difference is non-zero. However, we do not yet understand other parameters, for which $rank\left(A_{inter}\right)-rank\left(A^{\prime }\right)=0$, or the case of even *m*. Also, due to the first simplification, the concrete value of β is not found, and thus the concrete answer complexity of the resulting PIR as implied by Theorem 8 remains unknown.

Our current paper considers the case where $p_{1}=2$. We proceed by a quite straightforward generalization of [32]’s elimination process up until producing the matrix $A_{inter}$, except that we do not make the simplification of keeping a single row out of $M_{≢}$, but rather keep the entire matrix. The main divergence from [32] is that we do not perform the reduction modulo a subspace, but are able to directly check whether $rank\left(A_{inter}\right)-rank\left(A^{\prime }\right)>0$, and furthermore to compute the exact value of β. This is made possible, as the case where $p_{1}=2$ turns out to be particularly simple, and we managed to successfully analyze it directly (for all $p_{2},p$). The other cases (when *m* is odd, and *p* is not equal to $p_{1}$ or $p_{2}$) remain open.

## 2. Preliminaries

### 2.1. Some Notation

**Parameters of the secret sharing schemes.** Throughout this paper, we fix the notation for $p_{1}$, $p_{2}$ and *p* being prime numbers such that $p_{1}\neq p_{2}$, and $m=p_{1}\cdot p_{2}$ are the parameters of the secret sharing schemes and conversion. Later, considering the corner case $p_{1}=2$ in Section 3.4, we introduce the odd prime number *q* to set $p_{2}=q$.

**Matrices and block-matrices.** In this paper, we will consider matrices and block-matrices over a finite field $F=Z_{p}$. Those matrices are defined for 3 levels. The level-2 (“big”) block-matrices we denote by letter *A* with correspondent indexes. The elements of level-2 matrices are level-1 block-matrices which we denote by the letter *R* with a lower index equal to the upper index of its “host” *A*. The level-0 “small” matrices are square matrices, initially having the size m×m. For them, we use distinct letters.

For entry i,j of some matrix *X*, we use the standard notation of X[i,j]. Addressing the elements of level-2 and level-1 matrices, we address their blocks. Such, $A^{1}\left[i,j\right]$ denotes the block in the *i*th row and *j*th column of $A^{1}$. For level-0 matrices, we address the particular elements of this matrix. More generally, for a matrix $X\in F^{u\times v}$, for the subsets R⊆[v] of rows and C⊆[u] of columns, X[R,C] denotes the sub-matrix with rows (or block-rows) restricted to *R* and columns (or block-columns) restricted to *C*. Those rows and columns are ordered in the original order in *X*. As special cases, using a single index *i* instead of *R* (*C*) refers to a single row (column). A “·” instead of *R* (*C*) stands for [u] ([v]). Most of the time, index arithmetic will be done modulo the matrices’ number of (block-)rows and columns (we will however state this explicitly).

When we consider the case $p_{1}=2$ in Section 3.4, the level-1 matrices $R_{j}^{i}$’s are quite small and have only 2 level-0 blocks. Therefore, we omit the level-1 and address to level-0 blocks as to the entries of level-2 matrices $A^{(k),\ell }$.

**Some concrete matrices and vectors.** By the letter *I*, we denote the m×m identity matrix. If the identity matrix has a different size, we write this size down in the lower index. For instance, $I_{q}$ is a q×q identity matrix. By $a^{b\times c}$ we denote a b×c matrix with all elements equal to *a*. In case when a=0 and the size of this zero-block is clear from the context, we omit b×c and write 0 instead of $0^{b\times c}$. By $a^{b}$ we denote the row of *a*’s of the length *b*. For example, $1^{m}$ means the *m*-long string of 1’s.

By $e_{i}$ we denote the unity vector. The length of this vector is, as a rule, clear from the context, or it is specified in the accompanying text. The lower index specifies the position of 1 in this vector. In Section 2.5 and subsequently in Section 3.2, when we construct matrices in the basis $B=B_{1}\cup B_{2}$, the unity vectors have double indexes $e_{b,c}$. As explained in Section 2.5, there is the telescopic indexing system, and this double index points to the single position in the vector.

**Concatenation and circular shifts over matrices.** For matrices X,Y with the same number of columns, (X;Y) denotes the matrix comprised by concatenating *Y* below *X*. For matrices X,Y with the same number of rows, we denote by (X|Y) the matrix obtained by concatenating *Y* to the right of *X*.

In Section 3.4, we obtain the set of circularly shifted matrices. By $X^{<<k}$ we denote the matrix *X* with the circular left shift by *k* positions.

### 2.2. Secret Sharing Schemes

A *secret sharing scheme* is defined by pair of algorithms Sh=(Share,Dec). The randomized algorithm Share randomly splits a secret message s∈S into an *n*-tuple of shares, $\left(s_{1},\ldots ,s_{n}\right)$. The deterministic algorithm Dec reconstructs *s* from some allowed (*qualified*) subset of the shares. The set of all the qualified sets is called an *access structure* of the secret-sharing scheme. We say that Sh is *t*-private, and has a threshold access structure if any *t* shares jointly reveal no information about the secret *s*.

We say that Sh is linear over some finite Abelian ring *G* if S⊆G and each share $s_{i}$ is obtained by applying a linear function over *G* to the vector $\left(s,r_{1},\ldots ,r_{\ell }\right)\in G^{\ell +1}$, where $r_{1}$,…, $r_{\ell }$ are random and independent elements of *G*. A useful property of such schemes is that they allow evaluating locally linear functions of the shares such that additions and multiplications by the constant from *G*. In this work, we consider two types of linear secret sharing schemes:**Additive secret sharing:** the algorithm Share splits s∈G into *n* random ring elements that add up to *s*; the algorithm Dec reconstructs *s* by adding up all the shares. This scheme is (n−1)-private. Within the limits of this work, we consider a 3-additive scheme, where n=3.**CNF secret sharing:** the algorithm Share first splits s∈G into nt additive shares $s_{T}$, each labeled by a distinct set T∈[n]t, and then lets each share $s_{i}$ be the subset of $s_{T}$ apart from i∈T. For (2,3)-CNF we consider in this work, each of 3 parties obtains 2 additive shares out of 3, such that if additive shares of *s* are (a,b,c), then $s_{1}=\left(b,c\right)$, $s_{2}=\left(a,c\right)$, and $s_{3}=\left(a,b\right)$. This scheme is 1-private, as any two parties can sum their shares up to calculate the secret *s*.

See [33] for a survey on secret sharing.

### 2.3. Share Conversion

We recall the definition of (generalized) share conversion schemes as considered in our paper. Our definition is exactly the definition in [13], in turn, adopted from previous work.

**Definition** **1**([13])**.**
*Let $Sh_{1}$ and $Sh_{2}$ be two n-party secret-sharing schemes over the domains of secrets $S_{1}$ and $S_{2}$, respectively, and let $C\subseteq S_{1}\times S_{2}$ be a relation such that, for every $a\in S_{1}$, there exists at least one $b\in S_{2}$ such that (a,b)∈C. A share conversion scheme $convert(s_{1},\ldots ,s_{n})$ from $Sh_{1}$ to $Sh_{2}$ with respect to relation C is specified by (deterministic) local conversion functions $g_{1},\ldots ,g_{n}$ such that: If $\left(s_{1},\ldots s_{n}\right)$ is a valid sharing for some secret s in $Sh_{1}$, then $g_{1}\left(s_{1}\right),\ldots g_{n}\left(s_{n}\right)$ is a valid sharing for some secret $s^{\prime }$ in $Sh_{2}$ such that $(s,s^{\prime })\in C$.*

For a pair of Abelian groups $G_{1}$, $G_{2}$ (When $G_{1},G_{2}$ are rings, we consider $G_{1},G_{2}$ as groups with respect to the $“{+}^{”}$ operation of the rings), we define the relation $C_{S}$ as in [13].

**Definition** **2**(The relation $C_{S}$ [13])**.**
*Let $G_{1}$ and $G_{2}$ be finite Abelian groups, and let $S\subseteq G_{1}\backslash \left\{0\right\}$. The relation $C_{S}$ converts $s=0\in G_{1}$ to any nonzero $s^{\prime }\in G_{2}$ and every s∈S to $s^{\prime }=0$. There is no requirement when s∉S∪0. Formally,*
$$C_{S}=\left\{(s,0)|s\in S\right\}\cup \left\{\left(0,s^{\prime }\right):s^{\prime }\in G_{2}\backslash \left\{0\right\}\right\}\cup \left\{\left(s,s^{\prime }\right)|s\notin S\cup \left\{0\right\},s^{\prime }\in G_{2}\right\}$$

Given $m=p_{1}\cdot p_{2}$, where $p_{1}\neq p_{2}$ are primes and *p* is a prime, we consider pairs of rings $G_{1}=Z_{m},G_{2}=Z_{p}^{\beta }$. We denote the set a relation $C_{S_{m}}$ in this work is built with as $S_{m}=\left\{x\in G_{1}|\forall i\in \left[2\right],x\;\mathrm{mod}\;p_{i}\in \left\{0,1\right\}\right\}\backslash \left\{0\right\}$. I.e., $S_{m}=\left\{{\left(0,1\right)}_{Z_{m}},{\left(1,0\right)}_{Z_{m}},{\left(1,1\right)}_{Z_{m}}\right\}$, where ${\left(a,b\right)}_{Z_{m}}$ means the element of $Z_{m}$ which has the remainder *a* modulo $p_{1}$, and *b* modulo $p_{2}$. For $S=S_{m}$, we refer to $S_{m}$ as the *canonical relation* for $Z_{m}$.

### 2.4. The Characterization of BIKO

In Beimel et al. [13], $Sh_{1}$ is a 3-additive secret sharing scheme over $Z_{m}$, and $Sh_{2}$ is (2,3)-CNF sharing over $Z_{p}^{\beta }$. The conversion with respect to relation $C_{S_{m}}$ from $Sh_{1}$ to $Sh_{2}$ is considered. In [13], is proven that such a conversion exists iff a certain condition is satisfied by the matrix $M_{\equiv ,≢}$ over $Z_{p}$.

In matrix $M_{\equiv ,≢}$, the rows are indexed by triples $\left(a,b,c\right)\in Z_{m}^{3}$, corresponding to (2,3)-CNF sharings of some $s\in S_{m}\cup \left\{0\right\}$. The rows corresponding to s≠0 (i.e., to $s\in S_{m}$) form the upper part of the matrix, denoted by $M_{\equiv }$, and the rows corresponding to s=0 form the lower part, denoted by $M_{≢}$. In this way, $M_{\equiv ,≢}=\left(M_{\equiv };M_{≢}\right)$. The columns of $M_{\equiv ,≢}$ are indexed by values in $\left[3\right]\times Z_{m}\times Z_{m}$. Intuitively, an index (i,x,y) of a column corresponds to share $s_{i}$ (of *i*th server) of the (2,3)-CNF scheme being equal to (x,y). Rows are indexes by triples (a,b,c)=(s−b−c,b,c). There are *m* possible values for *a*, *b* and *c*, and 4 possible values for *s*. For a given *b* and *c*, and for a given *s* there is only one possible value of s−b−c, hence we replace the first index by simply *s*, and the matrix has $4m^{2}$ rows. The row indexed by (s,b,c) has 1 in the column (i,x,y), if the 3-additive shares (s−b−c,b,c) are agree with CNF-shares, and 0 otherwise. Thus, there are 1’s in cells [(s,b,c);(1,b,c)], [(s,b,c);(2,s−b−c,c)] and [(s,b,c);(3,s−b−c,b)], and 0’s elsewhere in this row.

The work in [13] provided a quantitative lower bound on β, depending on the degree difference between $M_{\equiv }$ and $M_{≢}$.

**Theorem** **8**(Theorem 4.5 [13])**.**
*Let $\beta =rank_{F_{p}}\left(M_{\equiv ,≢}\right)-rank_{F_{p}}\left(M_{\equiv }\right)$. Then, we have:*
*If β=0, then there is no conversion from (2,3)-CNF sharing over $Z_{m}$ to additive sharing over $Z_{p}^{\kappa }$ with respect to $C_{S_{m}}$, for every κ>0.**If β>0, then there is a conversion from (2,3)-CNF sharing over $Z_{m}$ to additive sharing over $Z_{p}^{\beta }$ with respect to $C_{S_{m}}$. Furthermore, in this case, every row v of $M_{≢}$ is not spanned by the rows of $M_{\equiv }$.*

Theorem 8 provides a full characterization via a condition that given $\left(p_{1},p_{2},p\right)$ can be verified in polynomial time in $\left(p_{1},p_{2},\log\left(p\right)\right)$. More precisely, the size of our matrix $M_{\equiv ,≢}$ is $4m^{2}\times 3m^{2}$, so verifying the condition amounts to solving a set of linear equations, which naïvely takes about $O(m^{6})$ time, or slightly better using improved algorithms for matrix multiplication, and the running time cannot be better than $\tilde{\Omega }\left(m^{4}\right)$ using generic matrix multiplication algorithms. Thus, the complexity of verification grows very fast with *m*, becoming essentially infeasible for $p_{1},p_{2}$ circa 50.

### 2.5. Our Starting Point—The Result of Paskin-Cherniavsky and Schmerler

The work [32] is made within BIKO’s setting. Starting with the matrix $M_{\equiv ,≢}$ they performed the sequence of elimination steps, according to the following lemma.

**Lemma** **2.**
*Let A denote a matrix in $Z_{p}^{v\times u}$, and let b=A[v,[u]]. Let $I_{1}\subseteq \left[v-1\right],I_{2}\subseteq \left[u\right]$ denote non-empty sets of rows and columns, respectively. $A^{\prime }$ is obtained from A by a sequence of row operations on A, so that $A^{\prime }\left[I_{1},I_{2}\right]$ is a basis of $A^{\prime }\left[\left[v\right],I_{2}\right]$, and the rest of the rows in $A^{\prime }\left[I_{1},I_{2}\right]$ are zero. Let $b^{\prime }=A^{\prime }\left[v,\left[u\right]\right]$. Then, $Rows(A^{\prime }\left[\left[v\right]\backslash I_{1},\left[u\right]\backslash I_{2}\right])$ span $b^{\prime }\left[\left[u\right]\backslash I_{2}\right]$ iff Rows(A[[v−1],[u]]) span b.*


In fact, the result of [32] is the proof of the existence of the conversion for finite rings $G_{1}=Z_{p_{1}p_{2}}$ to $G_{2}=Z_{p_{1}}^{\beta }$ with distinct odd $p_{1}$ and $p_{2}$, for which it was enough to prove that the first row of $M_{≢}$ is not spanned by $M_{\equiv }$. Therefore, the matrix considered in [32] contained the full matrix $M_{\equiv }$ and the single row from $M_{≢}$. (As in our work we solve the problem for 2 sets of parameters proving both positive and negative results, we consider the full $M_{\equiv ,≢}$ matrix). After two elimination steps which cut the matrix $M_{\equiv }$ to $m+m^{2}$ rows, and the permutation of columns, they introduced a new basis $B=B_{1}\cup B_{2}$, where
(1)$$\begin{array}{cc} B_{1}=\left\{-e_{b,i}+e_{b,i+1}\right. & \left.|b\in Z_{m},\;i\in \left\{0,...,p_{2}-2\right\}\right\},\\ B_{2}=\left\{e_{b,i+j\cdot {\left(1,0\right)}_{Z_{m}}}-e_{b,i+\left(j+1\right)\cdot {\left(1,0\right)}_{Z_{m}}}\right. & \left.|b\in Z_{m},\;i\in Z_{p_{2}},j\in Z_{p_{1}}\backslash \left\{p_{1}-1\right\}\right\}, \end{array}$$
where $e_{x,y}$ is a vector of length $m^{2}$ having 1 in the position indexed by (x,y) and 0’s elsewhere. Indexes *x* and *y* are taken modulo *m*.

In this new basis, the matrix $M_{\equiv ,≢}$ has the block structure and is separated into 3 “types” (layers): Type-1 and Type-2 layers compose $M_{\equiv }$, and Type-3 is $M_{≢}$ after several elimination steps and basis change. For the Type-1 matrix, the basic block is the *m*-component vector (1,…,1) and $p_{1}\times \left(p_{1}-1\right)$ block matrices $R_{1}^{2}$ and $R_{1}^{3}$ made from *m*-long vectors:(2)
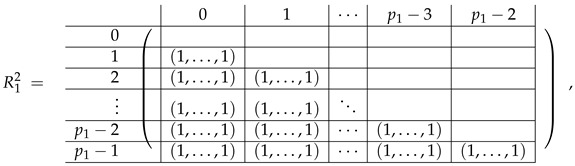

(3)
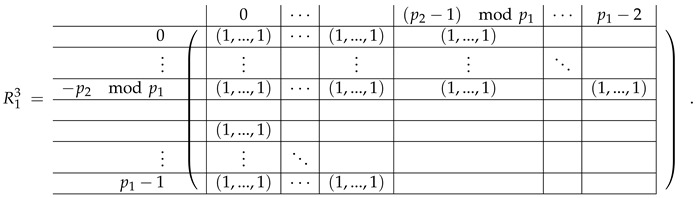


Basic m×m blocks of Type-2 are identity matrix *I*, and
(4)
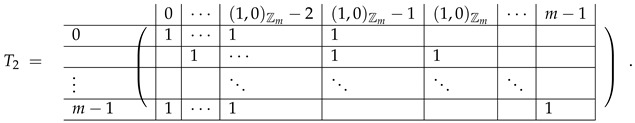


Bigger blocks composed from them have the size $p_{1}\times \left(p_{1}-1\right)$ “small” m×m blocks:
(5)
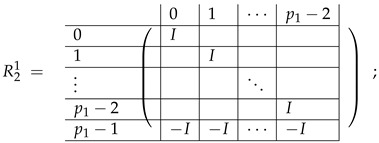

(6)
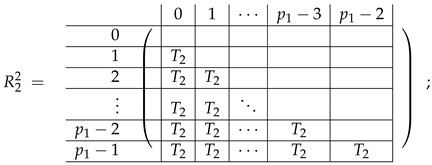

(7)
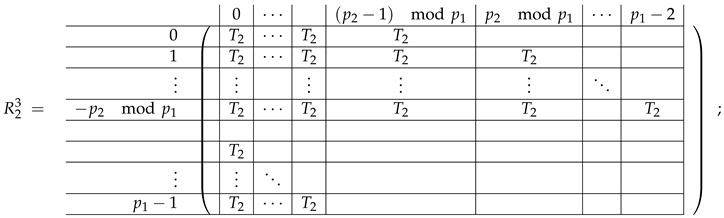

(8)
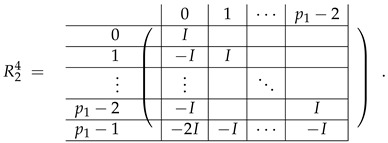


The matrix $M_{\equiv }$ is brought to the form $\left(A^{1};A^{2}\right)$, where each of matrices $A^{1}$ and $A^{2}$ are block matrices having the left and right parts (In [32], the matrices we are talking about have the additional upper index (6), which is omitted here. Thus, $A^{i}=A^{(6),i}=\left(A^{(6),L,i}|A^{(6),R,i}\right)$ for i∈1,2):(9)$$\left(A^{1};A^{2}\right),\;A^{1}=\left(A^{L,1}|A^{R,1}\right),\;A^{2}=\left(A^{L,2}|A^{R,2}\right),\;$$
where

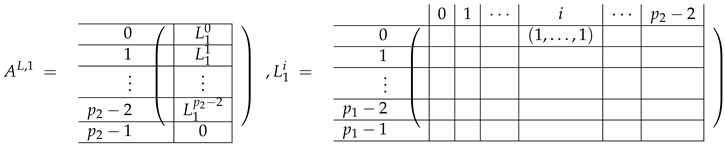

(10)
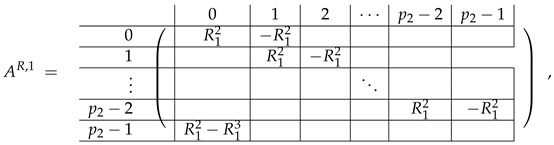


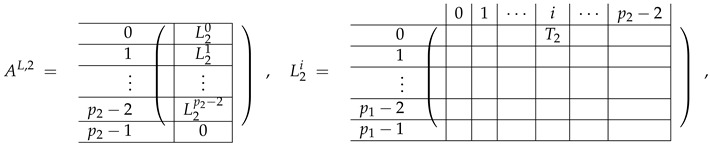

(11)
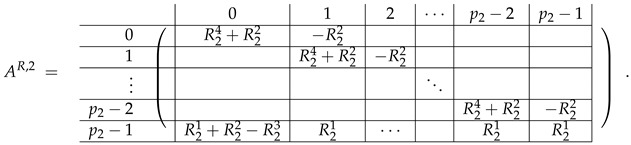


We remark that the appearance of $A^{R,1}$ and $A^{R,2}$ are slightly different from those in the work of Paskin-Cherniavsky and Schmerler. The difference is in $p_{2}-1$’st block-row and comes from the computational mistake made in [32] while changing the basis to *B*. We correct this mistake and bring the corrigendum in Appendix A.

In [32], only the first row from the Type-3 layer was constructed. For our purposes of obtaining β, we need the full Type-3 matrix. Therefore, we write here down the general formula for Type-3 rows taken from [32], and we use it in our next work to construct the entire Type-3 matrix in the basis *B*:(12)$$e_{\left(B,C\right)}-e_{\left(B,C+{\left(0,1\right)}_{Z_{m}}\right)}-\sum_{k=0}^{{\left(-1,0\right)}_{Z_{m}}}\left(e_{B+k,C}-e_{B+k,C+1}\right),\;\mathrm{where}\;B,\;C\in Z_{m}.$$

## 3. Our Result

### 3.1. Starting Point and Main Technical Tool

Our goal is to compute the difference between ranks of matrices $M_{\equiv }$ and $M_{\equiv ,≢}$. We start from the matrix $M_{\equiv }$ brought to the form (Equation 9) and we also construct the result of initial elimination steps over the matrix $M_{≢}$ from (Equation 12), obtained in [32]. We continue the process of elimination using Lemma 2 considering the case m=2q, where *q* is an odd prime number.

### 3.2. Construction of Type-3 matrix

First, as we need to compute the rank of the matrix $M_{\equiv ,≢}$, it is not enough to consider only a single row from the Type-3 matrix (which is the result of the sequence of elimination steps over $M_{≢}$. Therefore, our first step is to reconstruct this matrix from (Equation 12) and to perform initial elimination steps similar to those were made over matrix $A^{2}$ in [32] to bring it to the form (Equation 11).

To be consistent, we denote the initial Type-3 matrix as $A^{(-1),3}$, the intermediate result of the inner elimination steps over this matrix as $A^{(0),3}$, and the final result (on the same stage as (Equation 9)) as $A^{3}$. Next we describe the process of obtaining Type-3 matrix from (Equation 12). Recall that each of matrices *A* is separated in the left and right parts, where the left part contains indexes of vectors from basis $B_{1}$, and right—from $B_{2}$. Each row of $A^{(-1),3}$ is indexed by i,j,b such that the largest blocks are indexed by $i\in Z_{p_{2}}$, and contains blocks indexes by $j\in Z_{p_{1}}$. The smallest blocks indexed by $b\in Z_{m}$ are, in turn, parts of middle-size blocks.

First, rewrite (Equation 12) as
(13)$$\begin{array}{c} e_{\left(B,C\right)}-e_{\left(B,C+{\left(0,1\right)}_{Z_{m}}\right)}-e_{\left(B,C\right)}+e_{\left(B,C+1\right)}-\sum_{k=1}^{{\left(-1,0\right)}_{Z_{m}}}\left(e_{B+k,C}-e_{B+k,C+1}\right)=\\ =e_{\left(B,C+1\right)}-e_{\left(B,C+1-{\left(1,0\right)}_{Z_{m}}\right)}-\sum_{k=1}^{{\left(-1,0\right)}_{Z_{m}}}\left(e_{B+k,C}-e_{B+k,C+1}\right). \end{array}$$
Consider the case when $i\neq p_{2}-1$, j≠0. Each row indexed by (i,j,b) is determined by (Equation 13), where B=b, $C=i+j{\left(1,0\right)}_{Z_{m}}$. Then the first two terms in (Equation 13) are
(14)$$e_{B,C+1}-e_{B,C+1-{\left(1,0\right)}_{Z_{m}}}=e_{b,\left(i+1\right)+j{\left(1,0\right)}_{Z_{m}}}-e_{B,\left(i+1\right)+\left(j-1\right){\left(1,0\right)}_{Z_{m}}}=B_{2}\left[\left(i+1\right),\left(j-1\right),b\right].$$The term in the sum in (Equation 13) is
(15)$$\begin{array}{c} e_{B+k,C}-e_{B+k,C+1}=e_{b+k,i}-e_{b+k,i}+e_{b+k,i+{\left(1,0\right)}_{Z_{m}}}-...+e_{b+k,i+j{\left(1,0\right)}_{Z_{m}}}-\\ -e_{b+k,i+1}+e_{b+k,i+1}-e_{b+k,i+1+{\left(1,0\right)}_{Z_{m}}}-...-e_{b+k,i+1+j{\left(1,0\right)}_{Z_{m}}}=\\ =B_{1}\left[i,\left(b+k\right)\right]+\sum_{\ell =0}^{j-1}\left(B_{2}\left[i,\ell ,b+k\right]-B_{2}\left[\left(i+1\right),\ell ,b+k\right]\right). \end{array}$$When j=0, $i\neq p_{2}-1$ the sum in (Equation 15) turns to 0. As for (Equation 14), the first two terms in (Equation 13) are:
(16)$$e_{B,C+1}-e_{B,C+1-{\left(1,0\right)}_{Z_{m}}}=e_{b,i+1}-e_{B,i-{\left(1,0\right)}_{Z_{m}}}=-\sum_{\ell =0}^{p_{1}-2}B_{2}\left[\left(i+1\right),\ell ,b\right].$$When j≠0, $i=p_{2}-1$, the first two terms of (Equation 13) are the same as in (Equation 14). The only difference is the sum of terms:
(17)$$\begin{array}{c} e_{B+k,C}-e_{B+k,C+1}=e_{b+k,\left(p_{2}-1\right)+j{\left(1,0\right)}_{Z_{m}}}-e_{b+k,p_{2}+j{\left(1,0\right)}_{Z_{m}}}=\\ =-e_{b+k,0}+e_{b+k,p_{2}-1}-\sum_{\ell =0}^{p_{2}-1+j}\left(e_{b+k,\ell {\left(1,0\right)}_{Z_{m}}}-e_{b+k,\left(\ell +1\right){\left(1,0\right)}_{Z_{m}}}\right)+\\ +\sum_{\ell =0}^{j-1}\left(e_{b+k,\left(p_{2}-1\right)+\ell {\left(1,0\right)}_{Z_{m}}}-e_{b+k,\left(p_{2}-1\right)+\left(\ell +1\right){\left(1,0\right)}_{Z_{m}}}\right)=\\ =-\sum_{\ell =0}^{p_{2}-2}B_{1}\left[\ell ,b+k\right]-\sum_{\ell =0}^{p_{2}-1+j}B_{2}\left[0,\ell ,b+k\right]+\sum_{\ell =0}^{j-1}B_{2}\left[\left(p_{2}-1\right),\ell ,b+k\right]. \end{array}$$Finally, for $i=p_{2}-1$, j=0, the first two terms in (Equation 13) are according (Equation 16), and the terms in sum are as in (Equation 17), except from the term $\sum_{\ell =0}^{j-1}B_{2}\left[\left(p_{2}-1\right),\ell ,b+k\right]$, i.e.,
(18)$$-\sum_{\ell =0}^{p_{2}-2}B_{1}\left[\ell ,b+k\right]-\sum_{\ell =0}^{p_{2}-1+j}B_{2}\left[0,\ell ,b+k\right].$$

Substituting expressions (Equation 14)–(Equation 18) to (Equation 13) for appropriate i,j,b, we obtain the matrix which has the structure similar to $A^{2}$:(19)$$A^{(-1),3}=\left(A^{(-1),L,3}|A^{(-1),R,3}\right),$$
where

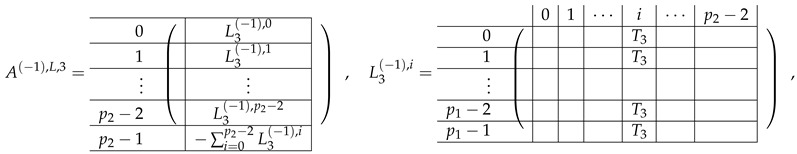


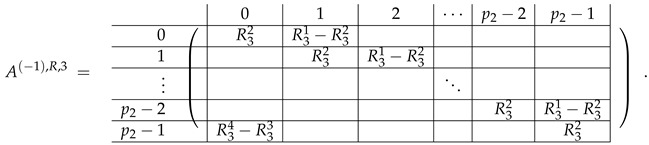


Here block matrices $R_{3}^{2}$ and $R_{3}^{3}$ are of the same form as $R_{2}^{2}$ (Equation 6) and $R_{2}^{3}$ (Equation 7) respectively, where the blocks $T_{2}$ are replaced with the blocks $T_{3}$:
(20)
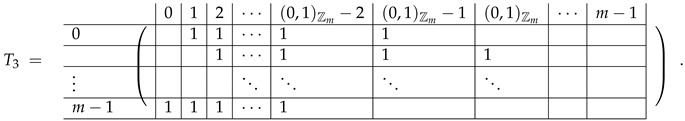


The matrix $R_{3}^{1}$ is similar to $R_{2}^{1}$, but with the opposite sign, and permuted rows: (21)
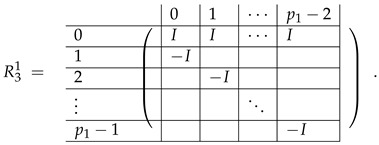

The matrix $R_{3}^{4}$ can be obtained from $R_{3}^{1}$ with the circular permutation of rows:
(22)
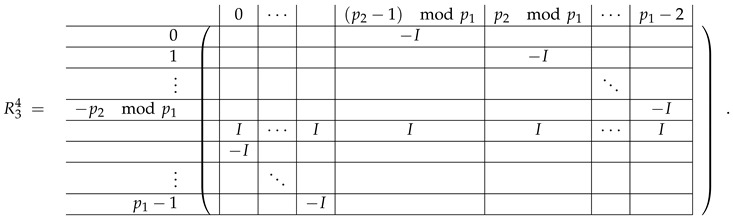


### 3.3. Elimination Steps in Type-3 Matrix

Following the way in [32] for elimination steps in $A^{2}$, we first sum the block-rows in (Equation 19) with ordinal numbers from 0 to $p_{1}-2$ to the last block-row. The resulting matrix $A^{(0),3}$ equals $A^{(-1),3}$ except the last row, where $A^{(0),L,3}\left[p_{2}-1,\cdot \right]$ is 0-block, and
$$A^{(0),R,3}\left[p_{2}-1,\cdot \right]=\left(\;R_{3}^{2}+R_{3}^{4}-R_{3}^{3}\;|\;R_{3}^{1}\;\left|\;...\;\right|\;R_{3}^{1}\;\right).$$

The second elimination step is an inner step in every block-row except from the last one (as those block-rows are the same in $A^{(-1),3}$ and $A^{(0),3}$, we can say that this step is performed over (Equation 19)). Namely, in any level-2 block-row $A^{(0),3}\left[i,\cdot \right]$ where $i\in \{0,...,p_{1}-2\}$, we subtract the level-1 block-row with the ordinal number 0 from all other sub-rows in this block-row. As a result, $A^{3}=\left(A^{L,3}|A^{R,3}\right)$, where the left-side matrix takes the form
(23)
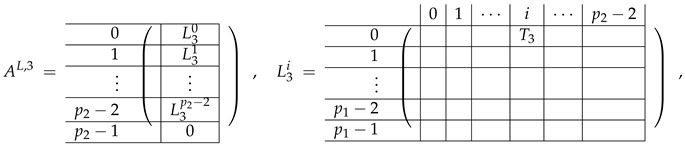

and the right-side matrix is
(24)
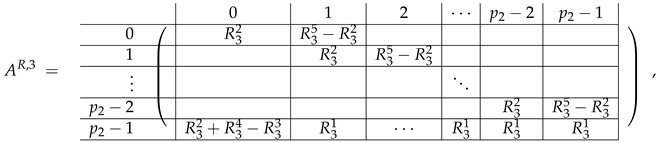

where
(25)
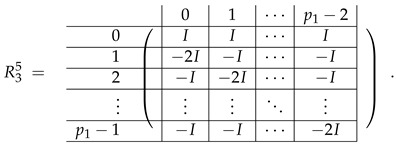


### 3.4. The Case of the Even *m* ($p_{1}=2$, $p_{2}=q$)

In this section, we consider the case $p_{1}=2$, $p_{2}>2$ for both p=2 and p>2 (including the case $p_{2}=p$). We obtain the feasibility results and, moreover, in the case of p=2 when the conversion from (2,3)-CNF over $Z_{2p_{2}}$ to three-additive secret sharing over $Z_{2}^{\beta }$ (as we prove) exists, we also compute β. We adopt the following *notation* in this section: $p_{2}=q>2$, and *p* are prime numbers, and m=2q (later we split this case into subcases p=2 and p>2). Our starting point is the block matrix $A=\left(A^{2};A^{1};A^{3}\right)$ over $Z_{p}$, where $A^{1}$ and $A^{2}$ are described in (Equation 9), and $A^{3}$ in (Equation 23) and (Equation 24).

We next consider the matrices in the case when m=2q, where *q* is an odd prime number. Below we write down the block matrices completing the matrix *A*. Each of the following matrices contains two square m×m blocks:(26)$$\begin{array}{cc} R_{2}^{1}=R_{3}^{1}=R_{3}^{4}=\left(\begin{array}{c} I\\ -I \end{array}\right) & ;\;R_{2}^{4}=R_{3}^{5}=\left(\begin{array}{c} I\\ -2I \end{array}\right);\\ R_{2}^{2}=\left(\begin{array}{c} 0\\ T_{2} \end{array}\right);\;R_{2}^{3}=\left(\begin{array}{c} T_{2}\\ 0 \end{array}\right) & ;\;R_{3}^{2}=\left(\begin{array}{c} 0\\ T_{3} \end{array}\right);\;R_{3}^{3}=\left(\begin{array}{c} T_{3}\\ 0 \end{array}\right). \end{array}$$

We would like to remind that $1^{m}=\left(1,...,1\right)$ is an *m*-element vector of 1’s, also $1^{q}$ is the *q*-element vector of 1’s. *I* is a m×m identity matrix, and $I_{q}$ is a q×q identity matrix.

Taking into account (Equation 26), the block matrix $A=\left(A^{2};A^{1};A^{3}\right)$ according to (Equation 9), (Equation 23) and (Equation 24) is the following:
(27)
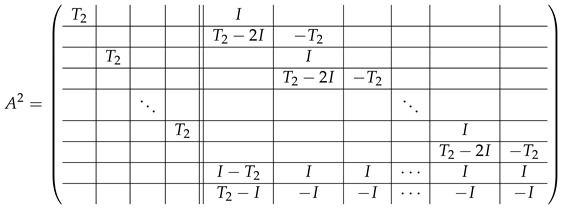

(28)
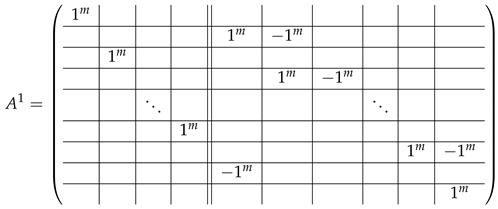

(29)
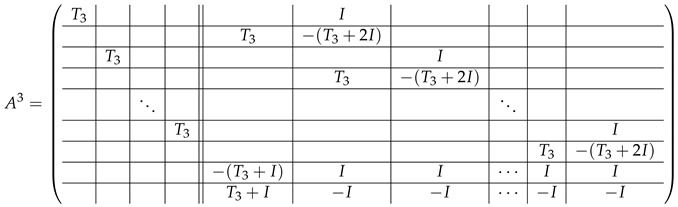


The left-side and the right-side matrices are divided by the double vertical line. All the subsequent matrices which we will obtain from *A* will have the additional upper indexes: $A^{\left(i\right)}=\left(A^{(i),2};A^{(i),1};A^{(i),3}\right)$, where *i* is the ordinal number of the matrix in the sequence of transformation steps. Within the limits of this section, we consider matrices $A^{(\cdot ),\ell }$ as if they have level-0 blocks as the entries, and we, therefore, address to level-0 blocks as to the entries of level-2 matrices $A^{(\cdot ),\ell }$.

First of all, we subtract $A^{2}$ from $A^{3}$ and rewrite all the matrices such that first go all the block-rows for $i<p_{2}-1$, j=0, and then the block-rows for $i<p_{2}-1$, j=1 and two last block-rows remain where they were before.

Matrices $T_{2}$ and $T_{3}$ depend on values ${\left(0,1\right)}_{Z_{m}}=q+1$ and ${\left(1,0\right)}_{Z_{m}}=q$. We introduce the new block $T=T_{3}-T_{2}$ and note that with respect to (Equation 26),
(30)
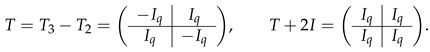

(31)
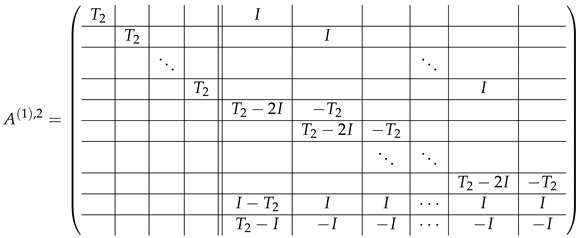

(32)
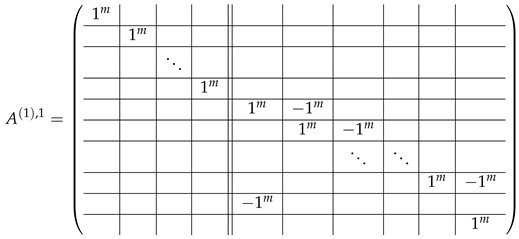

(33)
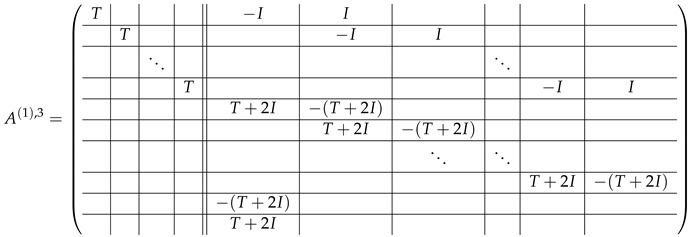


We note that the last block-row in $A^{(1),2}$ is the same as the previous one up to sign. The same observation we can make about $A^{(1),3}$.

#### 3.4.1. Internal Transformations in Matrices on the Level-2

We made some quite obvious steps inside each of the matrices $A^{(1),\cdot }$ to reach a more comfortable form. In $A^{(1),2}$ we eliminate the last block-row. In addition, we add all the block-rows from $A^{(1),2}\left[i\right]$ for i∈{q−1,...,2q−3} to block-row $A^{(1),2}\left[2q-2\right]$ and change the sign of this row. The resulting matrix is
(34)
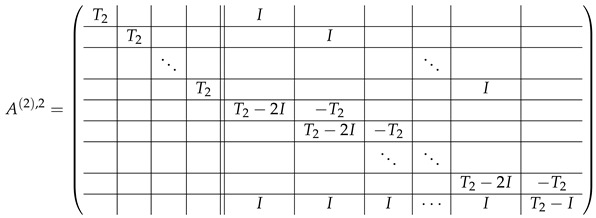


In $A^{(1),1}$, we change the sign of the pre-last row, move it to the position $p_{2}-1=q-1$ and make telescopic elimination by the following sequence of steps: starting with the row i=q, we subtract the previous row from *i*th, then change the sign of the *i*th row. Then we increment *i* by 1. We repeat this algorithm up to the last row of this matrix. Note, that the last row turns to 0, so we eliminate it from the matrix. The result is the matrix with (2q−1) rows:
(35)
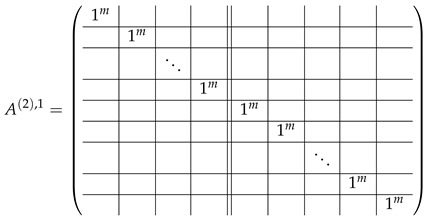


The same transformation we perform in $A^{(2),3}$, working with block-rows instead of rows. The result is the following matrix with (2q−1) block-rows:
(36)
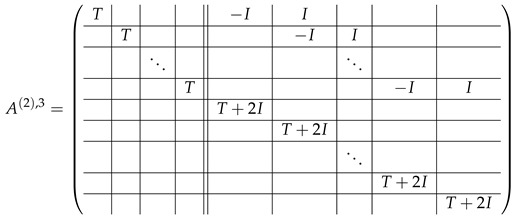


#### 3.4.2. Resolution of the Level-0 Blocks in $A^{(2),3}$

Consider the block-rows of the appearance $\left(\begin{array}{ccc} \cdots | & T+2I & |\cdots \end{array}\right)$ in (Equation 36). According (Equation 30), it equals to the matrix 
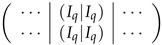
, where first *q* rows are the same as *q* last ones, and those *q* rows can be eliminated from the matrix. According to our notation, the q×m-block equal to the concatenation of two $I_{q}$’s is denoted as $\left(I_{q}|I_{q}\right)$. The rows we just considered belong to the local basis of $A^{(2),3}$.

Then we consider the block-rows $\left(\begin{array}{cccccc} \cdots & \left|T\right| & \cdots & \left|-I\right| & I| & \cdots \end{array}\right)$ in (Equation 36). Subtracting from this block-row the basis vectors from two corresponding block-rows $\left(\begin{array}{ccc} \cdots & \left|(I_{q}|I_{q})\right| & \cdots \end{array}\right)$ and taking into account (Equation 30), we transform it to:



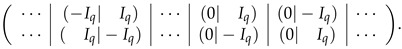



Again, the first *q* rows in this block-row can be crossed out from the matrix, as they are the same as *q* following ones up to the sign. Then the resulting matrix contains only q(2q−1) rows of the local basis:
(37)
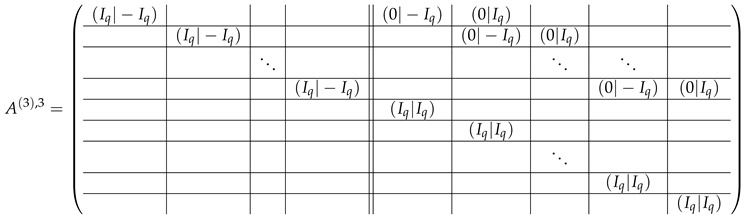


#### 3.4.3. Resolution of the Level-0 Blocks in ($A^{(2),2};A^{(2),1}$)

First, we consider block-rows of appearance $\left(\begin{array}{ccccc} \cdots & \left|T_{2}\right| & \cdots & \left|I\right| & \cdots \end{array}\right)$ in $A^{(2),2}$ (Equation (Equation 34)). For each j>q, $T_{2}\left[j\right]=T_{2}\left[0\right]+T_{2}\left[q\right]-T\left[j-q\right]$, hence we can bring all the $T_{2}\left[j\right]$ to 0 by the inner linear operations of rows in the block-row, which result in the following appearance of the block-row under the consideration:
(38)
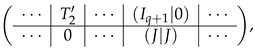

where $T_{2}^{\prime }=T_{2}\left[\left\{0,...,q\right\},\cdot \right]$, and *J* is a (q−1)×q-size matrix:
(39)
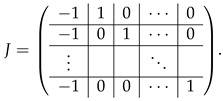


The set of all the upper block-rows as in (Equation 38) is in the echelon form and belongs to the basis of our matrix. Now, we pay attention to q−1 first rows in $A^{(2),1}$ (Equation (Equation 35)), taking into account that $1^{m}=T_{2}^{\prime }\left[0\right]+T_{2}^{\prime }\left[q\right]$. Subtracting the correspondent rows of $A^{(2),2}$ from $A^{(2),1}$, we bring the rows of the appearance $A^{(2),1}\left[i,\cdot \right]=\left(\begin{array}{cccc} \cdots & \left|1^{m}\right| & \cdots & ||\cdots \end{array}\right)$ to the form $\left(\begin{array}{cccc} 0|| & \cdots & \left|\left(-e_{0}|-e_{0}\right)\right| & \cdots \end{array}\right)$. Considering this resulting row together with the second block-row from (Equation 38), we can see that those are easily transformable to the form $\left(\begin{array}{cccc} 0|| & \cdots & \left|\left(I_{q}|I_{q}\right)\right| & \cdots \end{array}\right)$. As block-rows of the appearance $\left(\begin{array}{cccc} 0|| & \cdots & \left|\left(I_{q}|I_{q}\right)\right| & \cdots \end{array}\right)$ are composed from transformed rows of both matrices $A^{(2),2}$ and $A^{(2),1}$, we denote this merged matrix of q−1 block-rows as $A_{q-1}^{(3),1\&2}$:
(40)
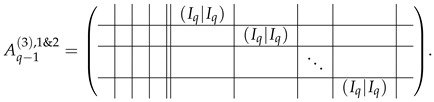

This matrix is in the left echelon form and thus consisted from the basis vectors of $M_{\equiv ,≢}$.

In addition, we consider the last block-row of $A^{(2),2}$, namely, $\left(\begin{array}{cccccc} 0|| & I| & I| & \cdots & \left|I\right| & T_{2}-I \end{array}\right)$ and subtract from there all the correspondent basis vectors of (Equation 40). Each block *I* then turns to 
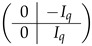
. Adding the upper half of the resulting block-row to the lower (and changing the sign of the upper one), we get 
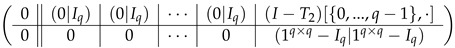
. We subtract the last row of $A^{(2),1}$ from each of last *q* rows of this block-row. Then these last *q* rows come to the form $\left(\begin{array}{ccc} 0|| & \cdots & |\left(I_{q}|I_{q}\right) \end{array}\right)$, and we append them to (Equation 40) to complete the matrix $A^{(3),1\&2}$. Then, subtracting the appropriate rows of the block $\left(I_{q}|I_{q}\right)$ from the block $\left(I-T_{2}\right)\left[0,...,q-1\right]$, we transform it to the form $\left(0^{q\times q}|I_{q}-N\right)$, where *N* is a *q*-dimension matrix
(41)
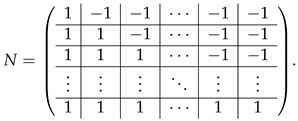


Finally, we consider the block-rows in $A^{(2),2}$ in (Equation 34) of the appearance $\left(\begin{array}{ccccc} 0|| & \cdots & \left|T_{2}-2I\right| & -T_{2}| & \cdots \end{array}\right)$. Let us consider the result of the elimination the basis $\left(I_{q}|I_{q}\right)$ from the block
(42)
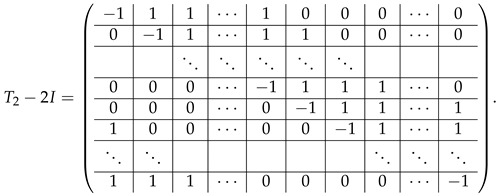

The result of the elimination is 
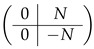
. For the block $-T_{2}$, the result of the elimination is 
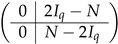
. Thus, each block-row under the consideration turns to the following *q*-row block-row $\left(\begin{array}{ccccc} 0|| & \cdots & \left|\left(0|N\right)\right| & \left(0|2I_{q}-N\right)| & \cdots \end{array}\right)$.

Thus, the result of all the transformations over $\left(A^{(2),2};A^{(2),1}\right)$ above is $\left(A^{(3),2};A^{(3),1\&2}\right)$, where

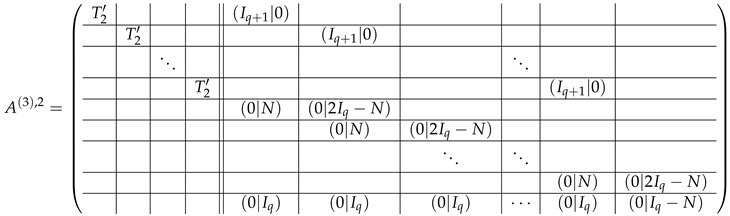

(43)
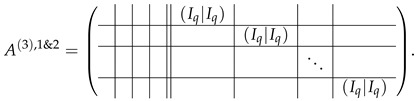


We note, that all the rows of $A^{(2),1}$ in (Equation 35) of the appearance $\left(\begin{array}{cccc} 0|| & \cdots & \left|1^{m}\right| & \cdots \end{array}\right)$ are spanned by $A^{(3),1\&2}$ as the row’s sums over the correspondent block-row.

#### 3.4.4. Resolution of the $A^{(3),1\&2}$ Basis

To apply Lemma 2, it is necessary to subtract vectors of basis $A^{(3),1\&2}$ from the first (q−1) block-rows of $A^{(3),2}$. The matrix $A^{(3),3}$ contains exactly the same block-rows as $A^{(3),1\&2}$ which can be simply crossed out.

Subtracting block $\left(I_{q}|I_{q}\right)$ from block $\left(I_{q+1}|0\right)$, we obtain the latest in form 
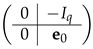
. We denote the (q+1)×q block $\left(\frac{-I_{q}}{e_{0}}\right)$ as $\left(-I_{q+1}^{\prime }\right)$. Then, after applying Lemma 2 to remove the basis $A^{(3),1\&2}$ and corresponding columns, we obtain the matrix
$$A^{\left(4\right)}=\left(A^{(4),2};A^{(4),3}\right),$$
where
(44)
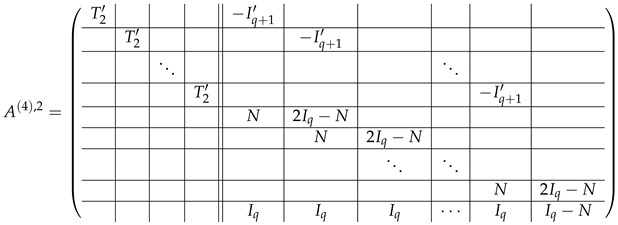


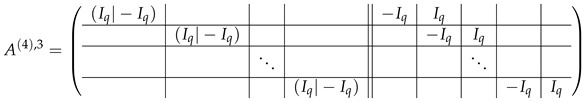


#### 3.4.5. Elimination of the Left-Side Matrices

We note that each row in block $\left(I_{q}|-I_{q}\right)$ is spanned by the rows of $T_{2}^{\prime }$, namely, $\left(I_{q}|-I_{q}\right)\left[j\right]=T_{2}^{\prime }\left[j\right]-T_{2}^{\prime }\left[j+1\right]$ (j∈{0,...,q−1}). Subtracting the correspondent rows of $A^{(4),2}$ from $A^{(4),3}$, and applying Lemma 2, we obtain

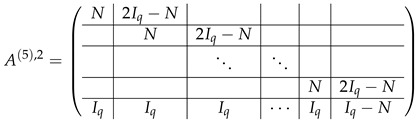

(45)
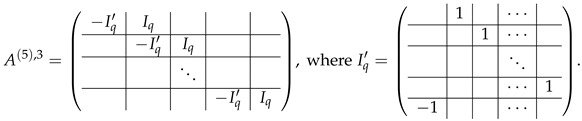

Here, the left side is crossed out, and matrix $A^{\left(5\right)}=\left(A^{(5),2};A^{(5),3}\right)$ is the matrix of q×q level-0 blocks.

#### 3.4.6. Resolution of *N* and $2I_{q}-N$ Blocks

Up to this moment, we performed transformations in matrices without connection to any particular modulus. Considering blocks *N* and $2I_{q}-N$ in $A^{(5),2}$ (Equation (Equation 45)), we can see two different situations taking into account the prime modulus p=2 or p>2:In the case of p>2, each *N* defined in (Equation 41) can be transformed to $I_{q}$ by the linear transformation steps such as additions of rows, multiplications by (−1) and $2^{-1}$ (which exists due to the fact that *p* is odd). Applying the same transformations to the adjacent block $2I_{q}-N$, we turn it into $-I_{q}^{*}$, where
(46)
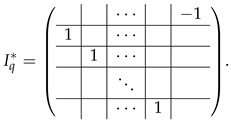
In the case of p=2, each block-row of the appearance $\left(\begin{array}{cccc} \cdots & \left|N\right| & 2I_{q}-N| & \cdots \end{array}\right)$ in (Equation 45) contains *q* equal rows $\left(\begin{array}{cccc} \cdots & \left|1^{q}\right| & 1^{q}| & \cdots \end{array}\right)$.

According to the dichotomy above, we next consider two cases.

#### 3.4.7. Case $p_{1}=2$, $p_{2}=q$, p>2

As described in the previous subsection, we start from matrix $A^{\left(6\right)}=\left(A^{(6),2};A^{(6),3}\right)$, where

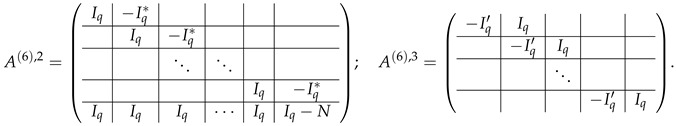


Performing inside each block-row of $A^{(6),3}$ two operations: multiplication rows at indexes from 0 to q−1 by (−1), and circular permutation of rows, we transform $A^{(6),3}$ to the form

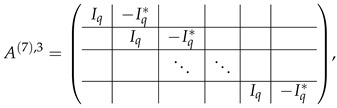

which is obviously spanned by rows of $A^{(6),2}$, and therefore $rank\left(M_{\equiv ,≢}\right)=rank\left(M_{\equiv }\right)$.

**Theorem** **9.**
*Assume m=2q, and p,q are odd prime numbers. Then there is no share conversion from (2,3)-CNF over $Z_{2q}$ to three-additive secret-sharing scheme over $Z_{p}^{\beta }$ for any β.*


**Proof.** The proof follows from Theorem 8 and the fact that $rank\left(M_{\equiv ,≢}\right)=rank\left(M_{\equiv }\right)$. □

#### 3.4.8. Case $p_{1}=2$, $p_{2}=q$, p=2

Here, we start from matrix $A^{\left(6\right)}=\left(A^{(6),2};A^{(6),3}\right)$, where

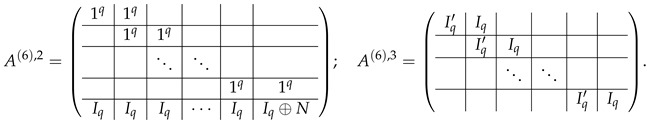


Performing the same permutation of rows in each block-row in $A^{(6),3}$ as for the case p>2, we obtain block-rows in form $\left(\begin{array}{cccc} \cdots & \left|I_{q}\right| & I_{q}^{<<1}| & \cdots \end{array}\right)$, where $I_{q}^{<<k}$ according to our notation is the result of the left *k*-bit circular shift in $I_{q}$. We remark that $I_{q}^{<<k_{1}}I_{q}^{<<k_{2}}=I_{q}^{<<(k_{1}+k_{2})\;\mathrm{mod}\;q}$. Subtracting the last block-row of $A^{(6),2}$ from the first block-row of $A^{(6),3}$, we obtain
$$A^{(7),3}\left[0\right]=\left(\begin{array}{cccccc} 0| & I_{q}⊕I_{q}^{<<1} & \left|I_{q}\right| & \cdots & \left|I_{q}\right| & I_{q}⊕N \end{array}\right).$$
The matrix $I_{q}⊕I_{q}^{<<1}$ is an invertible matrix, hence it is the linear transformation matrix. We subtract the row $\left(I_{q}⊕I_{q}^{<<1}\right)A^{(7),3}\left[1\right]$ from $A^{(7),3}\left[0\right]$ to obtain
$$A^{(7),3}\left[0\right]=\left(\begin{array}{cccccc} 0 & \left|0\right| & I_{q}⊕\left(I_{q}⊕I_{q}^{<<1}\right)I_{q}^{<<1} & |\cdots & \left|I_{q}\right| & I_{q}⊕N \end{array}\right).$$
We stress, that $I_{q}⊕\left(I_{q}⊕I_{q}^{<<1}\right)I_{q}^{<<1}=I_{q}⊕I_{q}^{<<1}⊕I_{q}^{<<2}$. Then we similarly subtract the 3rd block-row multiplied by the 3rd element of the first block-row, then 4th, and so on. As a result, the first block-row takes the form:$$A^{(7),3}\left[0\right]=\left(\begin{array}{cccccc} 0| & 0| & 0| & \cdots & \left|0\right| & I_{q}⊕N⊕I_{q}^{<<1}⊕I_{q}^{<<2}⊕\cdots ⊕I_{q}^{<<(q-1)} \end{array}\right).$$
Taking into account that ${⨁}_{k=0}^{q-1}I_{q}^{<<k}=1^{q\times q}=N$, the first block-row of $A^{(7),3}\left[0\right]$ equals zero and can be crossed out of the matrix.

Now, we make some elimination steps to bring matrix $A^{(6),2}$ to the echelon form (such that all the rows there are basis rows). For this, we subtract all the rows with even ordinal numbers from the first row of the last block-row. The resulting last block-row in $A^{(6),2}$ turns to $\left(\begin{array}{ccccc} K| & K| & \cdots | & K| & K⊕N \end{array}\right)$, where
(47)
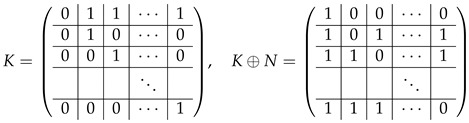

In both blocks *K* and K⊕N, the first row is spanned by others (as their sum), thus the first row of this block-row can be crossed out. The remaining rows in this block-row are basis vectors, which do not span $A^{(7),3}$, and thus, according to Lemma 2 can be thrown out the next consideration (together with the first row which also does not span $A^{(7),3}$). Then $A^{\left(7\right)}=\left(A^{(7),2};A^{(7),3}\right)$, where

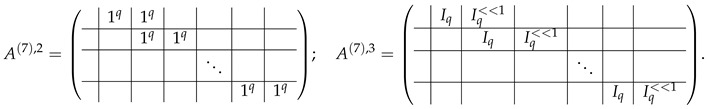


Matrix $A^{(7),3}$ contains (q−2) block-rows with *q* rows each. We make the last elimination step by subtracting each row of $A^{(7),2}$ from the first row of the correspondent block-row of $A^{(7),2}$. Then each block $I_{q}$ turns to *K*, and each $I_{q}^{<<1}$ turns to $K^{<<1}$, where the first row is the sum of others. Hence, each block-row in $A^{(7),3}$ loses the first row, and the rest of them are not spanned by the basis vectors of $A^{(7),2}$. Thereby, there are (q−2) block-rows with (q−1) rows each, and $rank\left(M_{\equiv ,≢}\right)-rank\left(M_{\equiv }\right)=\left(q-1\right)\left(q-2\right)$.

**Theorem** **10.**
*Assume m=2q, where q is an odd prime number. Then there exists a share conversion from (2,3)-CNF over $Z_{2q}$ to a three-additive secret-sharing scheme over $Z_{2}^{\left(q-1)(q-2\right)}$.*


**Proof.** The proof follows from Theorem 8 and $rank\left(M_{\equiv ,≢}\right)-rank\left(M_{\equiv }\right)=\left(q-1\right)\left(q-2\right)$. □

## 4. Computer Search Results on the Set $S_{m}$ and the Extended Set $S_{m}^{\prime }$

Table 4 in the work of Beimel et al. [13] reports ranks of the matrices $M_{\equiv }$ and $M_{\equiv ,≢}$ for m=6, 10, 14, 15, 21, 35 and p=2, 5, 7, 11. Unfortunately, some of the data there go against the proven properties of those matrices. For instance, in [32] it was proven that there exists a share conversion in case $m=p_{1}\cdot p_{2}$ and $p=p_{1}$, where $p_{1}$ and $p_{2}$ are distinct odd primes. At the same time, Table 4 in [13] shows that for the case m=5·7 it holds that $rank\left(M_{\equiv }\right)=rank\left(M_{\equiv ,≢}\right)$ over $Z_{7}$, which means the absence of the conversion. Moreover, $rank(M_{\equiv })$ cannot be less than $m^{2}$, since the matrix $M_{\equiv }$ has an identity block matrix of the size $m^{2}\times m^{2}$ in the upper left corner. However, for case m=35 in Table 4 in [13], this rank appears to be less. Therefore, we recalculated this table (also by computer search) in Table 1 and Table 2 to correct the result of [13] as well as to check the soundness of our derivations.

The results in Table 1 confirm our conclusions. Indeed, for the case of m=2q there is a conversion if and only if the modulus of the group is 2, and in this case β=(q−1)(q−2). Table 2 is relevant to the result of [32]. For the case of odd $m=p_{1}\cdot p_{2}$, there is a conversion if the modulus of the group *p* equals either $p_{1}$ or $p_{2}$.

In this paper, as well as in [32], only the case of the set $S_{m}$ of size 3 was considered. However, as we noted in the Introduction, the larger sets if the conversion for them exists, could result in MV families with higher VC dimension and hence in better PIR. For cases when the conversion exists in respect to the relation $C_{S_{m}}$, we also decided to check the extended sets $S_{m}^{\prime }$, trying different additional values from $Z_{m}$ and checking the ranks of $M_{\equiv }$ and $M_{\equiv ,≢}$.

For even *m*, we only tested possible extensions for $S_{m}$ modulo 2 (because if there is no conversion for a set $S_{m}$, then there is no conversion for any extended set). Of all the cases in Table 1, only for m=2·7 there are extended sets $S_{m}^{\prime }=\left\{1,3,7,8\right\}$ and {1,5,7,8} with β>0 (namely, β=6). The set $S_{m}^{\prime }=\left\{1,3,5,7,8\right\}$ provides β=0 and therefore the absence of the conversion.

Surprisingly, for odd *m*’s the result is more encouraging: for all *m* and *p* in Table 2 which provide β>0 for $S_{m}$, there were also extended sets $S_{m}^{\prime }$ with non-zero β. We summed them in Table 3. In the row $S_{m}^{\prime }\backslash S_{m}$, there is a subset extending $S_{m}$ up to $S_{m}^{\prime }$. It is interesting that *any* number of entries added from $S_{m}^{\prime }\backslash S_{m}$ to $S_{m}$ gives the same $rank(M_{\equiv })$ and β. It is also interesting that the set $S_{m}^{\prime }$ in all the checked cases with the odd *m* contains all the entries which are equal to 1 modulo $p_{2}$ (taking $p_{1}=p$) except from 1 and ${\left(0,1\right)}_{Z_{m}}$ which are already in $S_{m}$. Namely, $S_{m}^{\prime }\backslash S_{m}=\left\{{\left(2,1\right)}_{Z_{m}},...,{\left(p-1,1\right)}_{Z_{m}}\right\}$.

## Figures and Tables

**Table 1 entropy-24-00497-t001:** Rank of $M_{\equiv }$ and difference between ranks $M_{\equiv }$ and $M_{\equiv ,≢}$ ($rank(M_{\equiv })$; β) for some even *m* over different $Z_{p}$.

*m*	2·3=6	2·5=10	2·7=14	2·11=22	2·13=26
$S_{m}$	{1,3,4}	{1,5,6}	{1,7,8}	{1,11,12}	{1,13,14}
p=2	87 ; 2	247 ; 12	487 ; 30	1207 ; 90	1687 ; 132
p=3	89 ; 0	259 ; 0	517 ; 0	1297 ; 0	1819 ; 0
p=5	89 ; 0	259 ; 0	517 ; 0	1297 ; 0	1819 ; 0
p=7	89 ; 0	259 ; 0	517 ; 0	1297 ; 0	1819 ; 0

**Table 2 entropy-24-00497-t002:** Rank of $M_{\equiv }$ and difference between ranks $M_{\equiv }$ and $M_{\equiv ,≢}$ ($rank(M_{\equiv })$; β) for some odd *m* over different $Z_{p}$.

*m*	3·5=15	3·7=21	3·11=33	5·7=35
$S_{m}$	{1,6,10}	{1,7,15}	{1,12,22}	{1,15,21}
p=2	617 ; 0	1229 ; 0	3077 ; 0	3529 ; 0
p=3	593 ; 24	1169 ; 60	2897 ; 180	3529 ; 0
p=5	609 ; 8	1229 ; 0	3077 ; 0	3409 ; 120
p=7	617 ; 0	1217 ; 12	3077 ; 0	3457 ; 72

**Table 3 entropy-24-00497-t003:** Extensions for some sets $S_{m}$ allowing the share conversion.

*p*	3	3	3	5	5	7	7
*m*	3·5=15	3·7=21	3·11=33	3·5=15	5·7=35	3·7=21	5·7=35
$S_{m}^{\prime }\backslash S_{m}$	{11}	{8}	{23}	{4,7,13}	{8,22,29}	{4,10,13,16,19}	{6,11,16,26,31}
$rank(M_{\equiv })$	607	1201	2989	627	3511	1257	3547
β	12	30	90	2	30	2	12

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
