# Peer review of "New Bounds and a Generalization for Share Conversion for 3-Server PIR"

_entropy, 2022, doi:10.3390/e24040497_

Round 1
Reviewer 1 Report
In this research article, using algebraic techniques from the previous work of Paskin-Cherniavsky and Schmerler in 2019, the authors prove the existence of the share conversion for specific parameters. Although the concrete efficiency of 3-server PIR is not improved, their result seems to be promising in a broader context of constructing PIR with three servers, according to the authors.
In my opinion, even this paper is concrete and organized. However, it is very technical in its present form. It is challenging for general readers outside the context of share conversion to acquire a simple overview of the paper. Most terminologies are not well-defined and described explicitly in the article, e.g., what is CNF?
Overall, I think this paper is a solid work. However, the report could be easier to follow if some key insights, essential procedures, and used arguments were clearly addressed. I would focus on the high-level idea here, and below I list some main comments.
1. What is the main novelty/contribution in terms of methodology compared to [31]? It seems that the methods used in this paper are very similar to [31]. Probably, the studied parameters are different. Perhaps the authors need to highlight the main differences between this work and [31] (except for the parameters). Or asking alternatively, what is the main challenge to solve the problem for the parameters studied in this paper?
2. There are still some minor typos that should be corrected throughout the paper. E.g.,
a. No “A” in “Abstract: A Private Information Retrieval (PIR) protocols, ….”
b. Line 491: there should be a space before “according to the following lemma.”
c. Line 623: “the larger sets, it the conversion for them exists, …” should be “if the conversion for them exists, ….”
Please consider proofreading the paper more carefully again.
3. Before the Preliminaries Section, it was hard to understand the terminology of “BIKO Framework” and “CNF.” it is suggested to use less technical terms in the Introduction section.
Author Response
We thank the reviewers for their useful and insightful comments.
We tried to take into account all your comments and suggestions to make the work better. We changed the storyline in our Introduction to make our motivation, conclusions and significance of our result for future work clearer. We added the subsection "1.5. Instantiations of BIKO and future directions of our work" to not confuse the background and the context of the work. Also, we added the necessary explanations to other subsections of the Introduction. In particular, new Remark 1 to the subsection "1.3 BIKO framework" gives a contrast and emphasizes the limitations of BIKO.
Following are some replies.
1. We now made the difference in techniques more clear in the "our techniques" section. In a nutshell, for p1=2, as opposed to odd p1,p2, we managed to perform direct and complete analysis of the ranks of the relevant pair of matrices M=, M=,!= , leading to full understanding of the setting with m=2p2 and some p. In [PS19], and the setting of odd m, a complete analysis was quite hard, and instead of deciding the condition that characterizes the existence of a conversion, we instead resolve a (possibly) more stringent condition, which is easier to resolve. Thereby, [PS19] leaves a subset of parameters unresolved, while for even m me exactly characterize when a share conversion exists. The case of odd m remains not fully resolved due to the technical difficulty in understanding the ranks of the matrices that still persists.
2. Fixed.
3. We added high level explanations of the relevant terms.
Reviewer 2 Report
This work focuses on share conversion, which has been shown to be useful for private information retrieval (PIR) - an important cryptography problem. While no improvements on the communication cost of PIR are reported, a deeper understanding of share conversion could be interesting by itself.
I wish to know more about the 'limits' of share conversion. Can all best known PIR protocols be understood in the framework of share conversion? Are we aware of any downside of share conversion based PIR protocols?
The majority of this work focuses on 3 server PIR or specific number of servers. How general is share conversion based PIR solutions? And please compare with the best known protocols.
The English can be much improved. For example,
Line 1: 'A' PIR protocol's'
Line 13: 'a' efficient
Line 24: BIKO is not defined/referred yet
Please carefully go over the whole paper.
Author Response
We thank the anonymous reviewers for their useful and insightful comments.
Some replies follow.
- Regrading limitations of the BIKO framework - we added a discussion that indeed some of the state of the art PIR protocols do not fall into this framework. In particular, the work of [DG14], which puts forward the best 2-server PIR follows an outline that can be seen as a generalization of the BIKO framework (see Remark 1 in the revised paper for a broader discussion). It demonstrates that it is unlikely that the (concrete version of) the BIKO framework as described in our work is not the only route to consider towards improving PIR complexity, can generalizing it in various directions may be useful. Nevertheless, it is a fruitful direction, with the advantage that PIR protocols with improved complexity can be found via a systematic search of certain share conversions between secret sharing schemes with finite domains.
- We added a clearer explanation regarding the generality of the framework, and added details on the particular structure of the framework. This particular framework captures all "3rd generation" (state of the art) PIR protocols we are aware of, except of the 2-server protocol of [DG14] as discussed above. Earlier protocols, as well as [DG14] are often close to it in spirit, but do not fit the concrete details of it as we described it.
- We also changed the storyline in our introduction to make our motivation, conclusions and significance of our result for the future work clearer.
Round 2
Reviewer 1 Report
The authors have addressed my comments precisely and properly. Thus, I recommend acceptance of the paper.
Reviewer 2 Report
My comments are sufficiently addressed.